# LANGUAGE-CONDITIONED MULTI-STYLE POLICIES WITH REINFORCEMENT LEARNING

## ABSTRACT

Recent studies have explored the application of large language models (LLMs) in language-conditioned reinforcement learning (LC-RL). These studies typically involve training RL agents to follow human instructions in domains such as object manipulation, navigation, or text-based environments. To extend these capabilities for following high-level and abstract language instructions with diverse style policies in complex environments, we propose a novel method called LCMSP, which can generate language-conditioned multi-style policies. LCMSP first trains a multi-style RL policy capable of achieving different meta-behaviors, which can be controlled by corresponding style parameters. Subsequently, LCMSP leverages the reasoning capabilities and common knowledge of LLMs to align language instructions with style parameters, thereby realizing language-controlled multi-style policies. Experiments conducted in various environments and with different types of instructions demonstrate that the proposed LCMSP is capable of understanding high-level abstract instructions and executing corresponding behavioral styles in complex environments. [1]

## 1 INTRODUCTION

Natural language serves as a bridge between agent behavior and human instructions. Training agents to follow natural language instructions has been a long-standing problem (Winograd, 1972). Recent research has explored this problem using language-conditioned reinforcement learning (LC-RL) (Luketina et al., 2019), demonstrating remarkable performance in tasks such as navigation, object manipulation, and arrangement (Tellex et al., 2011; Hill et al., 2020; Brohan et al., 2023; Pang et al., 2024; Szot et al., 2024).

Beyond these tasks, reinforcement learning (RL) has shown outstanding performance in more complex environments such as Dota 2 (Berner et al., 2019), StarCraft (Vinyals et al., 2019), Gran Turismo (Wurman et al., 2022), and Google Research Football (GRF) (Song et al., 2024). Despite the effectiveness of RL in these complex environments, controlling these RL policies to follow instructions and achieve specific desirable behaviors remains very challenging. This is because instructions in such scenarios can involve long-horizon planning, combinatorial action spaces, and require specific behavioral styles. For example, in the 5v5 scenario of the GRF environment, agents need to control five players over 3,000 steps per episode. The instruction "*Prioritize defensive duties and perform a quick counterattack when opportunities arise.*" involves team formation, division of offensive and defensive roles, and preferences on ball possession, passing, dribbling, scoring, etc. In such complex environments, human instructions can be high-level[2] and abstract, specifying not only task completion but also describing a behavioral style.

Recent LC-RL studies still cannot achieve RL policies in complex environments to follow high-level instructions. This is because training LC-RL often requires determining whether the instruction has been successfully executed as a form of reward. In tasks like navigation and object manipulation, rules can be used to judge whether an instruction is completed (Hill et al., 2020; Driess et al., 2023; Tan et al., 2024; Pang et al., 2024; Szot et al., 2024). However, for abstract instructions in complex

---

[1]The code can be found in the Supplementary Material.

[2]In this study, low-level instructions represent target instructions that can be completed by a few meta-behaviors, while high-level instructions are complex and require more meta-behaviors to participate.

environments, it is challenging to use rules to determine whether instructions are completed and if the agent's behavior aligns with the specified style. Some methods use human annotation to determine the success of each episode (Brohan et al., 2023) or employ expert data for inverse-RL to obtain rewards (Fu et al., 2019; Bahdanau et al., 2019), eliminating the need for predefined rules but requiring additional labor costs. Moreover, their environments still focus on simple navigation and object arrangement tasks. Apart from instruction evaluation, executing abstract instructions in complex environments still faces the following two challenges: understanding instructions and aligning them within the target environment, and executing the correct behaviors specified by the corresponding instructions in the environment.

In response to these challenges, we propose a novel LC-RL method named Language-Conditioned Multi-Style Policies (LCMSP). This method enables a single model to be trained with diverse behavioral styles that can be flexibly controlled by multi-style parameters. Before the training process, we design a series of meta-behaviors that encompass the main behaviors of the agent in the target environment. Then, a multi-style RL approach is designed to train a policy capable of executing meta-behaviors with different degrees and combinations. It also provides a controllable mechanism to switch between these styles (Mysore et al., 2022; Le Pelletier et al., 2021). The trained policy can follow diverse instructions without needing to evaluate instruction completion during training. During inference, by leveraging the language understanding capabilities of large language models (LLMs), language instructions are translated into corresponding multi-style parameters using a Degree-to-Parameter (DTP) prompting method. Figure 1 presents an overview of the inference process for the proposed method.

The proposed LCMSP endows agents with diverse behaviors in complex environments while achieving fine-grained control over RL policies through language instructions. Experiments conducted both in the autonomous driving environment Highway and across various scenarios in GRF demonstrate the method's robust instruction follow capabilities and high performance. Our main contributions are: (1) We propose a novel method that combines multi-style RL policies with LLMs to follow language instructions in complex environments; (2) Our method can accept high-level abstract language instructions, enabling not only task completion but also the expression of corresponding behavioral styles, with the degree of style adjustable at a fine-grained level; (3) Extensive experiments across multiple environments, various instruction types, and policy evaluations demonstrate the effectiveness and generality of this method, also showing insensitivity to different LLMs. [3]

## 2 RELATED WORKS

**Multi-Style Reinforcement Learning.** Multi-style RL methods aim to train a single policy model capable of exhibiting diverse behavioral styles, with applications in various scenarios including game AI (Mao et al., 2024; Mysore et al., 2022; Le Pelletier et al., 2021; Shen et al., 2020), robotic control (Abdolmaleki et al., 2020), autonomous driving (Zhang et al., 2023), and text generation (de Langis et al., 2024; Cho et al., 2022). Multi-objective RL (MORL) is an RL training framework that also attempts to generate policies with varying behaviors (Abdolmaleki et al., 2020; Mossalam et al., 2016). The goal of MORL is to learn policies that simultaneously optimize multiple competing objectives. Some research efforts focus on learning a set of policies to approximate the Pareto frontier (Pirotta et al., 2015) of optimal solutions (Zuluaga et al., 2016; Chen et al., 2019). Other approaches, such as Yang et al. (2019) and Basaklar et al. (2023), train a single preference-conditioned policy using vectorized variants of standard RL algorithms. The key difference between our proposed method and MORL is that MORL seeks optimal solutions on the Pareto frontier under a set of given objectives, whereas our method achieves different style policies by varying reward settings with their style parameters. Multi-task RL (MTRL) is another closely related approach to generating multi-style policies (Lan et al., 2024; Liu et al., 2021). MTRL trains a single model to complete a number of distinct tasks, each requiring specific skills, where each skill can be treated as a distinct behavioral style. Yang et al. (2020) and He et al. (2024) use a routing network that estimates different routing strategies to reconfigure the base network for each task. Sodhani et al. (2021) and Cho et al. (2022) learn a task embedding network in addition to the policy network, allowing for knowledge sharing across tasks. MTRL is commonly trained on limited independent tasks such as those in Meta-World (Yu et al., 2020). In contrast, our multi-style policy integrates diverse meta-behaviors

---

[3]Demonstration videos showing our multi-style policies are available at: `https://sourl.cn/vwgFMk`.

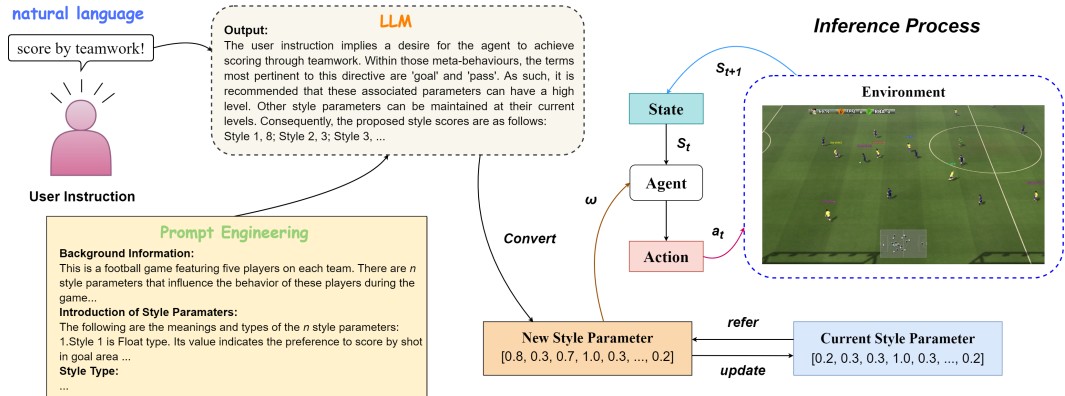

Figure 1: Overview of the inference process of LCMSP. When an instruction is provided to the system, the LLM combines the pre-prepared prompt with the instruction to generate a corresponding response. This response is then transformed into a style parameter, denoted as $\omega$. Upon receiving the current state $S_t$ and the style parameter $\omega$, the agent determines the appropriate action $a_t$, which is executed in the environment, leading to the next state $S_{t+1}$. Unless new instructions are received, the agent continues to operate using the last set of style parameters.

and style parameters, enabling combinations that produce a wide range of behaviors, rather than just completing a limited set of tasks.

**Language-Conditioned Reinforcement Learning.** Early works addressed the issue by parsing language instructions into semantic vectors that represent the structure of the instruction in relation to world entities (Tellex et al., 2011; Chen & Mooney, 2011). Recent approaches tend to embed both the instruction and observation to condition the policy (Jiang et al., 2019; Pang et al., 2024; Brohan et al., 2023; Hill et al., 2020; Song et al., 2023). Hill et al. (2020) encode natural language human instructions using BERT (Devlin et al., 2019), which are then fed into the policy. TALAR (Pang et al., 2024) develops a task language for policy training and a translator to convert natural language into the task language. SayCan (Brohan et al., 2023) grounds LLMs through value functions to select low-level language-conditioned policies. Most prior works focus on simple low-level instructions, such as object manipulation tasks like picking and placing objects (Pang et al., 2024; Hill et al., 2020; Jiang et al., 2019), or navigation tasks where the goal is to reach a specific entity (Tellex et al., 2011). Though some approaches combine object picking and navigation tasks to form long-horizon tasks (Brohan et al., 2023; Song et al., 2023), they remain sequential combinations of low-level instructions. In contrast, our method can understand and execute high-level abstract instructions by leveraging the knowledge of LLMs combined with multi-style RL policies.

**Large Language Models for RL.** LLMs exhibit exceptional natural language understanding and logical reasoning abilities (Zhao et al., 2023), which can facilitate downstream decision-making tasks in RL. In certain navigation and object manipulation tasks, some approaches leverage the reasoning capabilities of LLMs to generate high-level plans (Huang et al., 2022; Brohan et al., 2023; Song et al., 2023), while others directly utilize LLMs to output actions (Szot et al., 2024; Tan et al., 2024). For instance, LLaRP (Szot et al., 2024) appends fully-connected layers trained using RL after the LLM to output actions and value functions. TWOSOME (Tan et al., 2024) outputs corresponding action tokens in a text-based environment and fine-tunes the LLM. Additionally, several methods in robotic control environments use LLMs to generate rewards for RL training (Ma et al., 2024; Yu et al., 2023). Our approach integrates LLMs with multi-style RL policies. By leveraging the comprehension and reasoning capabilities of LLMs, we translate instructions with hidden meanings and varying degrees into corresponding style parameters, thereby fulfilling human intentions.

## 3 PRELIMINARIES

**Reinforcement Learning** is typically formulated as a Markov Decision Process (MDP), defined by the tuple $\langle S, A, P, r, \gamma \rangle$. Here, $S$ represents the state space, and $A$ denotes the action space. The

transition function $P : S \times A \times S \rightarrow [0,1]$ captures the environment dynamics, specifying the probability of transitioning to state $s_{t+1} \in S$ from state $s_t \in S$ by taking action $a \in A$. The reward function $r : S \times A \rightarrow R$ assigns a reward to each state-action pair. A policy $\pi(a|s)$ is the agent's behavior function, mapping states to actions or providing a probability distribution over actions. The value function $V^\pi(s)$ evaluates the quality of a state by predicting future rewards. In RL, the goal is to learn an optimal policy $\pi^*$ that maximizes the expected discounted sum of rewards. Formally, the optimal policy is defined as: $\pi^* = \arg\max \mathbb{E}_s [V^\pi(s)]$, where the value function $V^\pi(s)$ is given by: $V^\pi(s) = \mathbb{E}_{\tau \sim \pi, P(s)} [\sum_{t=0}^\infty \gamma^t r(s_t, a_t)]$. Here, $\gamma \in [0,1)$ is the discount factor, and $\tau \sim \pi$ with $P(s)$ indicates sampling a trajectory $\tau$ for a horizon $T$ starting from initial state $s_0$ using policy $\pi$, and $s_t \in \tau$ represents the state at $t$-th time step in the trajectory $\tau$.

**Conditioned RL** can be formulated as an augmented MDP, which is defined by the tuple $\langle S, C, A, P, r_c, \gamma \rangle$. The additional tuple element $C$ is the space of conditions, and other elements retain their definitions from the standard MDP. The reward function $r_c : S \times A \times C \rightarrow R$ assigns the reward to each state-action-condition triplet. Similarly, the policy $\pi(a|s, c)$ maps both states and conditions to actions. The objective in conditioned RL is to find a policy $\pi(a|s, c)$ that maximizes the expected discounted sum of rewards: $\mathbb{E}_{\tau \sim \pi, P(s), P_c(c)} [\sum_{t=0}^\infty \gamma^t r(s_t, a_t, c)]$, where $P_c(c)$ represents a distribution over conditions in $C$. This objective can also be expressed with a standard MDP by augmenting the state vector with a condition vector, explicitly conditioning the policy on $c$ allows the agent to adapt its behavior based on different conditions.

## 4 METHOD

### 4.1 REWARD SHAPING WITH META-BEHAVIOR

Before the training process, it is essential to define the agent's *meta-behaviors*, which represent the key behaviors the agent can perform in the training environment. A meta-behavior can signify an "objective", a "task", or even a "behavioral process" of the agent within the environment. The agent's preference for a particular meta-behavior indicates the frequency or degree to which the agent employs that meta-behavior. By appropriately combining meta-behaviors, the agent can deal with most conditions in the target environments. For instance, in the Highway scenario, an autonomous driving environment, we can design four meta-behaviors: *Speed*, *Time to Collision*, *Change Lane*, and *Lane Preference*. A human instruction like "*Increase speed, minimize lane changes*" relates to the *Speed* and *Change Lane* meta-behaviors in this environment. To achieve fine control over meta-behaviors, we perform reward shaping for each one, allowing the preference for a meta-behavior to be adjusted through a corresponding parameter in the reward function. For example, we can assign a reward parameter $\omega_1$ for speed and another reward parameter $\omega_2$ for changing lanes. When $\omega_1$ and $\omega_2$ are set to 1, the agent will frequently change lanes at high speed due to these high positive rewards. Conversely, when $\omega_1$ and $\omega_2$ are set to 0, the agent will have less preference for changing lanes or maintaining high speed because there are no benefits for these behaviors. The combination of preferences for various meta-behaviors formulates a "style" of policy. The parameters that control the preferences of meta-behaviors in the reward function are called "style parameters" in this study. We denote the style parameters as $\boldsymbol{\omega}$, which is a vector $[\omega_1, \omega_2, \ldots, \omega_n]$, where $\omega_i$ is the $i$-th parameter controlling the preference of the $i$-th meta-behavior. A specific set of $\boldsymbol{\omega}$ values can form a distinct policy. The design of meta-behaviors and their reward shaping processes are preparatory steps in the training of LCMSP, as shown in Figure 2. Details of these steps are provided in Appendix A.3.

### 4.2 MULTI-STYLE POLICY GENERATION

Style parameters play a crucial role in generating multi-style policies within a single model. These parameters, denoted as $\boldsymbol{\omega}$, can be viewed as conditions within the RL training process. For each training scenario, at the beginning of an episode, a set of style parameters is randomly generated by the style generator. Throughout the episode, the agent receives the state $s$ and style parameters $\boldsymbol{\omega}$ from the environment as inputs to produce actions. The reward obtained by the agent in this episode is modulated by the style parameters $\boldsymbol{\omega}$, enabling the agent to execute different policy styles under identical states due to varying style parameters.

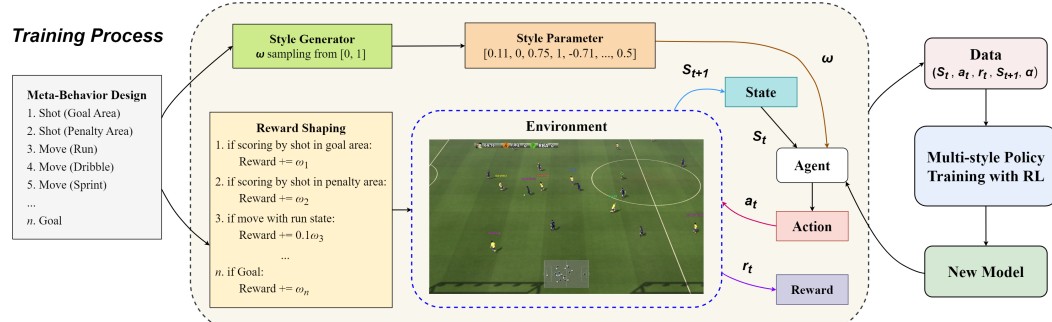

Figure 2: Overview of the training process of LCMSP. For each training scenario, we initially design the agent's meta-behaviors, followed by a reward shaping process for these meta-behaviors. This process establishes the relationship between the style parameters and the corresponding meta-behaviors. Before each episode, the style generator produces a set of style parameters. These parameters dictate the magnitude of the rewards obtained when the agent executes the associated meta-behaviors within the environment. Upon receiving the environment's state and style parameters, the agent get the corresponding action from the policy model and executes it. Subsequently, the environment provides the reward for the current action and the next state. These data is then fed into the RL algorithm for training, and through continuous iterations, new policies are generated.

Based on the categorization of RL algorithms, the style-conditioned policy optimization criteria can be divided into two classes. For RL algorithms optimized by policy gradient methods, the policy optimization criterion $J_{\pi_\theta}$ is proportional to the advantage function $A^{\pi_\theta}$, as shown below:

$$J_{\pi_\theta} \propto \log\left(\pi_\theta(a \mid s, \omega)\right) A^{\pi_\theta}(s, \omega, a) = \log\left(\pi_\theta(a \mid s, \omega)\right)\left(Q^{\pi_\theta}(s, \omega, a) - V^{\pi_\theta}(s, \omega)\right) \quad (1)$$

where the policy $\pi$ is parameterized by $\theta$, $V^{\pi_\theta}(s, \omega)$ is the value-function, and $Q^{\pi_\theta}(s, \omega, a)$ is the Q-function representing the expected return of taking action $a$ in state $s$ under style parameters $\boldsymbol{\omega}$.

The value estimator $V_\phi$, parameterized by $\phi$, is optimized with optimization criteria $J_{V_\phi^\pi}$:

$$J_{V_\phi^\pi} \propto \|\left(V_\phi^\pi(s, \omega) - \left(r^\omega(s, \pi(s, \omega)) + V_\phi^\pi(s', \omega)\right)\|\right. \quad (2)$$

where $s'$ is the next state obtained from the environment after taking action $a$, and $r^\omega$ is the reward function adjusted by style parameters $\boldsymbol{\omega}$.

For RL algorithms optimized by Q-value-based methods, the policy optimization criterion is proportional to the Q-value:

$$J_{\pi_\theta} \propto Q^{\pi_\theta}(s, \omega, a) = \mathbb{E}_{P(s')}\left[r^\omega(s, a) + \gamma Q^{\pi_\theta}(s', \omega, \pi(s', \omega))\right] \quad (3)$$

where $Q^{\pi_\theta}$ is parameterized by $\theta$, and $\gamma$ is the discount of the next Q-value.

The representation of style parameters can be learned through a style encoder. The encoded representations of the style parameters are then concatenated with the state features and forwarded to the subsequent network layers for processing. This process ultimately yields the actions and value/Q-values. When the values of the style parameters change, the state features remain unaffected. The policy change is produced by the subsequent network parameters, which facilitates the learning of both state and style features.

### 4.3 ALIGNING LANGUAGE INSTRUCTIONS WITH TARGET POLICIES

When applying trained multi-style policies, selecting an appropriate target behavior typically depends on the specific requirements of the instruction. For example, in autonomous driving, individuals in a hurry might opt for a policy favoring higher speeds, while those who prioritize safety may prefer a more cautious approach. However, the complexity of the environment and instructions often complicates the determination of a suitable target behavior. Utilizing LLMs to guide the application of multi-style policies in specific environments shows promise. Nonetheless, as the conditions in

multi-style policy training become increasingly complex and numerous, and as the behaviors specified by instructions grow more abstract, it becomes crucial for LLMs to understand both the instructions and the conditions. Additionally, generating style parameters directly from instructions poses a significant challenge for LLMs. This process requires understanding each parameter's meaning, identifying those relevant to the instruction, and determining their specific values.

To address these challenges, we propose a prompt method capable of interpreting the instruction and generating appropriate style parameters to guide the multi-style policy. We refer to this method as the Degree-to-Parameter prompt (DTP). The process of generating style parameters using DTP is illustrated in Figure 3. In this approach, background information introduces the application scenario and the concept of multi-style parameters to the LLM. These style parameters are categorized into two types: *Float* and *Bool*. A *Float*-type parameter is continuous, ranging from 0 to 1, whereas a *Bool*-type parameter is binary, taking values of either 0 or 1. The *Float*-type parameters are mapped to 11 preference degrees ranging from 0 to 10. Smaller values indicate lower preference, and larger values indicate higher preference; for instance, 0 and 10 can represent "unwilling" and "enthusiastic" preferences, respectively. The *Bool*-type parameters have two preference degrees: *deactivate* and *activate*. The introduction of style parameters provides their parameter types and controlled meta-behaviors in the scenario for all style parameters.

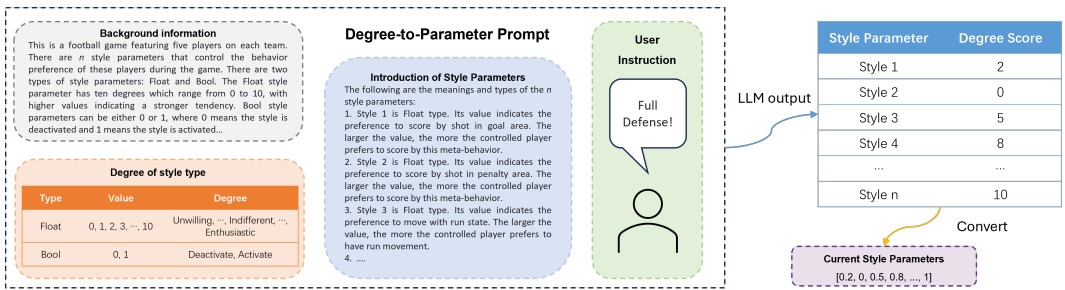

Figure 3: An illustration of the process for generating style parameters using DTP prompt engineering. The process consists of three fundamental modules: background information, an introduction of style parameters, and an introduction table of style types. For a given scenario, the contents of these modules remain constant. User instructions are integrated with these modules to form the complete prompt, which is then input into the LLM to obtain a style degree table. A new set of style parameters is then generated based on the style degree table.

The LLM identifies the meta-behaviors related to the instructions and presents the degree scores. For example, the instruction "*full defense!*" implies that the user wants the agent to fully defend. Consequently, meta-behaviors related to defense should have a high preference, while meta-behaviors related to offense should have a low preference. Finally, new style parameters can be automatically mapped based on the generated degree scores. An example about DTP method's prompt is provided in Appendix B.

The DTP method offers several advantages: 1) Ease of prompt design: It only requires explanations of the environment and meta-behaviors. 2) Generalization across environments: By leveraging the common sense and reasoning capabilities of LLMs, it adapts effectively to various environments. 3) Controllability and interpretability: Humans can observe the intermediate process in which the LLM aligns instructions with the relevant meta-behaviors, evaluating the degree of each style parameter. Moreover, if the behavior does not satisfy human expectations, the corresponding style parameters can be adjusted. 4) Robustness to LLM choice: Different LLMs demonstrate similar understanding of instructions, as illustrated in Figures 5 and 11.

## 5 EXPERIMENTS

To evaluate the proposed LCMSP method, we designed four scenarios based on two popular RL environments: Highway and GRF. In the GRF environment, we developed three scenarios involving single-player, two-player, and 5v5. Initially, we conducted experiments to test LCMSP's ability to follow low-level human instructions in the Highway environment and in the single-player and two-

player GRF scenarios. Subsequently, we evaluated LCMSP on the 5v5 scenario of GRF, a complex environment characterized by long-horizon, multi-agents, to demonstrate its capability in understanding and executing high-level human instructions. Additionally, we performed experiments to assess the method's ability to align instructions with style parameters as well as the diversity and controllability of multi-style policy behaviors.

## 5.1 Highway Environment

**Environment:** The Highway environment is a 2D top-down view autonomous driving simulation that supports multiple tasks. In this study, we utilize the `highway-v0` task, where the controlled vehicle navigates a multilane highway populated with other vehicles. The agent is responsible for controlling its speed and driving direction, aiming to reach a target speed while avoiding collisions. The primary meta-behaviors include *speed*, *time to collision (TTC)*, *lane change*, and *lane preference*. We design four rewards to regulate these meta-behaviors, which can be adjusted using style parameters. Additionally, a collision penalty is imposed on agents that collide with other vehicles. By selectively combining different styles of these four meta-behaviors, we derive distinct behavior styles to evaluate the LCMSP's ability of following human instructions. Detailed information about the scenario settings, state and action spaces, and reward shaping can be found in Appendix A.

**Instructions:** To thoroughly assess LCMSP's capability to comprehend and execute natural language instructions, we constructed five distinct instruction types, summarized in Table 1. The Normal, Long, and Short instruction types encompass all designed behavior styles, while the Unseen and Inference types cover only a subset. For each behavior style in the Normal, Long, Short, and Inference types, we generated ten instructions, and for the Unseen type, we generated 30 instructions per behavior style. To automate instruction generation, we first created several examples, designed a generation prompt, and then employed a LLM to generate the remaining instructions for all behavior styles in each instruction type. The Highway experiment utilized a total of 810 instructions. Detailed information on instruction generation and examples can be found in Appendices C and E.

Table 1: Explanation and examples of all instruction types.

| Instruction Type | Explanation | Instruction Examples |
|---|---|---|
| Normal | Describe the desired behavior set. | Let the car decelerate as it transitions to the slow lane. |
| Long | Incorporated lengthy instructions with additional adjectives and expressions. | Navigate into the right lane for a leisurely drive. With caution, and smoothly decelerate as you merge with the slower vehicles traveling there. |
| Short | Extremely concise instructions. | Right lane, slow driving. |
| Unseen | Unseen behavior styles during baseline method training. | While taking the wheel, slow your speed but not your lane changes. |
| Inference | Instructions that require inference without directly stating the behavior style. | My destination is calling me; I ought to make the journey brisker. |

**Results:** To demonstrate the instruction-following capability of the LCMSP method, we report both alignment accuracy and execution success rate. Alignment accuracy reflects the ability to match instructions with the ground truth style parameters. Execution success rate is assessed using criteria specifically designed for each behavior style within the environment, with details provided in Appendix D.1. As shown in Table 2, the LCMSP method achieves a high alignment accuracy of 91.4% in the Highway environment and correctly aligns most instructions requiring inference. The overall execution success rate is 88.8%, indicating effective execution of the corresponding instructions based on the aligned style parameters.

The performance of multi-style policies is crucial for execution success. Therefore, we tested the trained policies with different styles. Table 3 presents statistical indicators under various style parameters. In this experiment, we adjusted the target style parameters and recorded their related indicators, while other style parameters were randomly sampled. We observed that variations in each style parameter effectively influenced the corresponding indicators. For example, the agent's speed significantly increased as the style parameter for *Speed* became larger. This indicates that the multi-style RL training process can effectively produce policies exhibiting distinctly different styles, adjustable according to the meaning and extent of the style parameters. Furthermore, to demonstrate

Table 2: Alignment accuracy and execution success rates across instruction types

| Instruction Type | Alignment Accuracy | Execution Success |
|---|---|---|
| Normal | 97.4% ± 0.3% | 91.5% ± 1.6% |
| Long | 83.0% ± 0.5% | 92.4% ± 2.1% |
| Short | 96.2% ± 0.3% | 88.3% ± 3.7% |
| Inference | 73.3% ± 3.3% | 77.8% ± 3.5% |
| Unseen | 97.8% ± 1.1% | 81.9% ± 3.2% |
| Total | 91.4% ± 0.6% | 88.8% ± 0.7% |

the performance of multi-style policy, we report the execution success rates of behaviors corresponding to given style parameters, as shown in Appendix D.2. Detailed information on the multi-style training process can be found in Appendices A.4 and A.6.

Table 3: Indicator values for target style parameters across different settings

| Target parameters | Indicators | Indicator value at different style parameters | | |
|---|---|---|---|---|
| | | 0 / Deactivate | 0.5 | 1 / Activate |
| Speed | Speed value (m/s) | 19.6 ± 0.0 | 24.2 ± 0.3 | 28.9 ± 0.2 |
| TTC | TTC value (s) | 7.7 ± 0.9 | 7.8 ± 1.1 | 8.2 ± 1.1 |
| Change lane | Change action ratio (%) | 38.6 ± 5.7 | 39.3 ± 6.0 | 40.3 ± 6.0 |
| Lane preference (Left) | In left lane ratio (%) | 8.9 ± 9.5 | / | 33.1 ± 19.6 |
| Lane preference (Middle) | In middle lane ratio (%) | 60.0 ± 23.7 | / | 89.8 ± 3.0 |
| Lane preference (Right) | In right lane ratio (%) | 7.2 ± 7.7 | / | 28.3 ± 6.1 |

## 5.2 SINGLE-PLAYER AND TWO-PLAYER SCENARIOS IN THE GRF ENVIRONMENT

**Environment:** To verify LCMSP's ability to follow low-level instructions in the GRF environment, we design two simple scenarios: single-player and two-player. In these scenarios, the controlled team consists of one player and two players, respectively. For the single-player scenario, we apply four meta-behaviors: *Area X*, *Area Y*, *Move type*, and *Shot type*. For the two-player scenario, we apply five meta-behaviors: *Hold ball preference*, *Pass preference*, *Formation type*, *Shot type*, and *Move type*. These meta-behaviors reflect most actions in football and can be controlled by their corresponding style parameters. More details about these two scenarios are provided in Appendix A.

**Instructions:** The construction process of the instructions is similar to that in the Highway environment, resulting in 1,760 instructions for the single-player scenario and 620 for the two-player scenario. Generation details and examples are provided in Appendices C and E, respectively.

**Baselines:** We compare LCMSP with another LC-RL method, TALAR(Pang et al., 2024), which fine-tunes a BERT model(Devlin et al., 2019) as a translator to encode natural language instructions into inputs for RL policies. Additionally, we test LCMSP with three different LLMs: GPT-4o (OpenAI, 2024), Claude 3.5-Sonnet (Anthropic, 2024), and Llama 3.1-8b (Dubey et al., 2024). The implementation details of these baselines are given in Appendix D.3.

**Results:** Figure 4 shows the execution success rates of all baselines on five types of instructions. The results demonstrate that, for Normal and Short instructions, TALAR performs slightly worse than the LCMSP method. On the Unseen instruction sets, LCMSP significantly outperforms TALAR, which may be attributed to the excellent zero-shot capabilities of LLMs. TALAR also performs poorly on long instruction sets. This is due to the BERT model summarizing long sequences into a single [CLS] token. Moreover, the LCMSP method exhibits lower variance on inference instructions and demonstrates robustness across different LLMs with minimal performance differences.

## 5.3 5v5 SCENARIO IN GRF ENVIRONMENT

**Environment:** We designed ten meta-behaviors to train a multi-style policy in the 5v5 scenario: *Win Preference*, *Goal*, *Lose Goal*, *Hold Ball*, *Get Possession*, *Pass*, *Spacing*, *Shot*, *Move*, and *Formation*. Accordingly, ten style parameters and their corresponding reward functions are employed, details are provided in Appendix A.3. The multi-style policy in the 5v5 scenario is trained by competing against built-in AI and further improved through a self-play approach due to its adversarial nature.

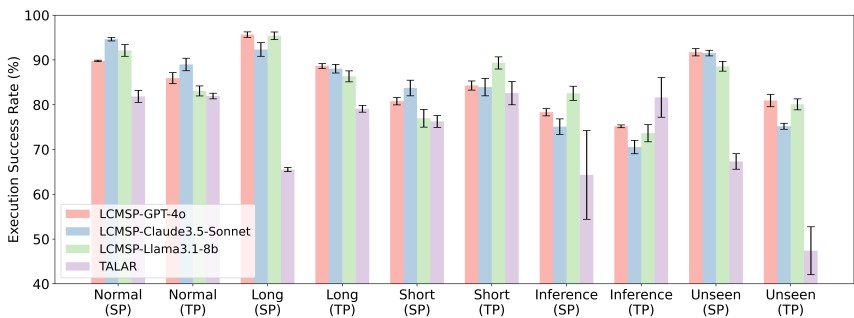

Figure 4: Execution success rates of all baselines on five types of instructions. SP and TP represent single player and two player scenarios, respectively.

More details regarding the training algorithms, hyperparameters, and configurations are provided in Appendices A.4 and A.6.

**Instructions:** To demonstrate that the LCMSP method can understand and execute abstract high-level instructions, we designed six tactics analogous to real-world football strategies: *Positive Attack*, *All-out Attack*, *Balanced Play*, *Counter Attack*, *Park the Bus*, and *Tiki-Taka*. For each tactic, we generated 30 natural language instructions, each articulated with a brief sentence to elucidate the corresponding strategy. Examples of instructions for each tactic are provided in Table 16.

**Results:** In the complex 5v5 scenario with high-level instructions, it is not feasible to use rules to determine whether the instructions have been successfully executed. We used in-game metrics to evaluate whether LCMSP behaves according to the specified tactics (see Figure 5). The number of goals and shot attempts for both the *Positive Attack* and *All-out Attack* tactics exceed those of the *Balanced Play* tactic. However, the *All-out Attack* tactic results in a lower win rate due to a higher number of conceded goals and fewer draws, attributable to its overly aggressive style. The *Counter Attack* tactic exhibits a higher draw rate; to facilitate counter-attacks, the formation is positioned deeper, with a larger space between forwards and defenders. The *Park the Bus* tactic features more compact spacing and results in a very high draw rate. The *Tiki-Taka* tactic boasts the highest possession ratio and number of pass attempts, with closer spacing facilitating short passes; however, its overly cautious approach results in fewer goals. These results indicate that the LCMSP method can accurately comprehend high-level instructions and execute corresponding behavior styles.

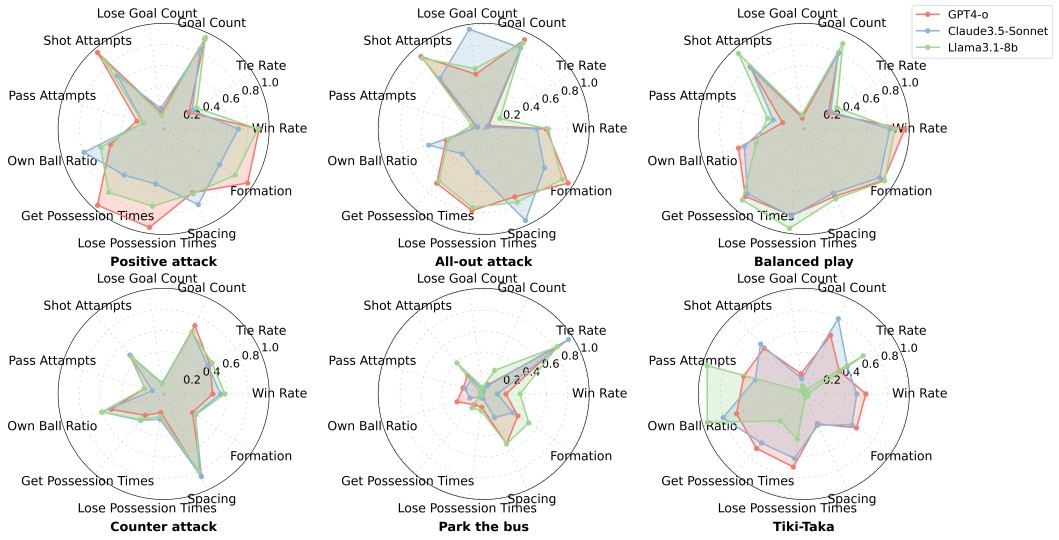

Figure 5: Comparison of in-game metrics under different tactical instructions. The metric values are normalized based on maximum and minimum values.

The performance across different LLMs is largely similar, though there are differences in the degree of meta-behavior in certain cases. For instance, Claude sacrifices more defense in the *All-out Attack* tactic, while Llama3.1 exhibits more extreme passing attempts in *Tiki-Taka*. Appendix D.4.1 offers a detailed analysis of the degree of style parameters across various tactics. Figure 6 illustrates examples of rendered frames during the execution of the *Counter Attack* and *Tiki-Taka* tactics, respectively. These visualizations demonstrate that the trained multi-style policy effectively executes the desired tactics.

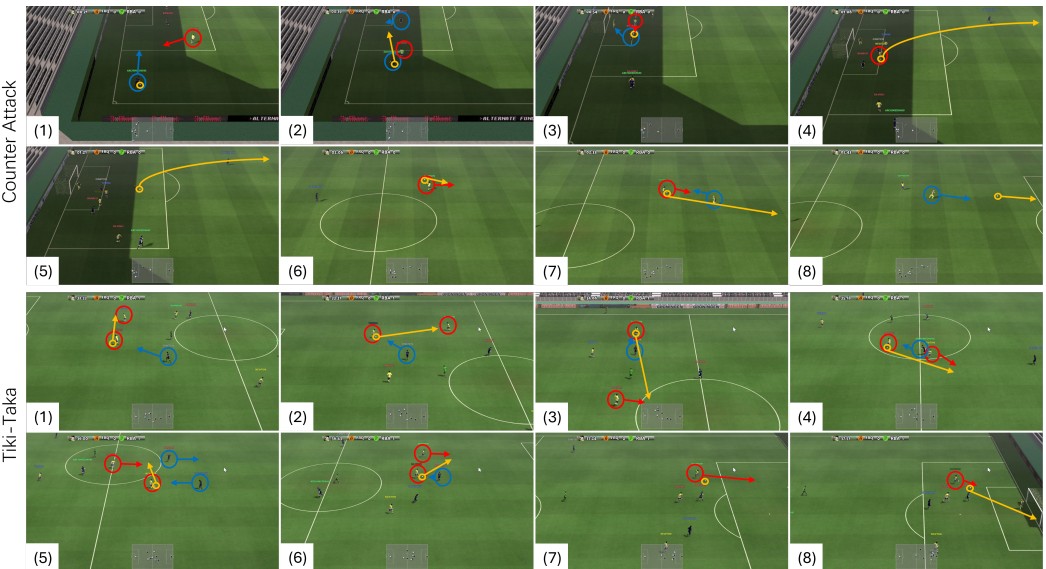

Figure 6: Rendered frames showcasing the *Counter Attack* and *Tiki-Taka* tactics. Red, blue, and yellow represent our team, the opposing team, and the ball, respectively. Circles denote players and the ball, while lines indicate movement and passing directions. As intended, the *Counter Attack* executes a long pass following a defensive interception in a deep position, while the *Tiki-Taka* tactic advances through compact spaces using a series of short passes.

Policies trained using the LCMSP method are capable of exhibiting multiple styles in complex environment, and their behavior can be finely adjusted through the corresponding style parameters. The variations in in-game metrics resulting from fine-grained adjustments of these style parameters are detailed in Appendix D.4.2. Furthermore, to better illustrate the representations of meta-behaviors under different style parameters, we present the in-game statistical metrics for extreme style parameters in Appendix D.4.3.

## 6 CONCLUSION AND DISCUSSION

In this study, we introduced the Language-Conditioned Multi-Style Policy (LCMSP), which enables agents to exhibit highly diverse behaviors with fine-grained control through natural language instructions. The policy is trained using a specially designed multi-style RL method, and alignment between natural language and multi-style policies is achieved via the Degree-to-Parameter (DTP) prompt with LLMs. Previous LC-RL methods have primarily focused on environments involving object manipulation and navigation. In contrast, LCMSP is capable of executing abstract, high-level instructions in complex environments. Experiments across multiple environments and instruction types demonstrate the effectiveness and generality of our method. A limitation of LCMSP is its long inference time, exceeding one second, primarily due to the alignment process with LLMs, despite our relatively small RL model. As LLMs continue to evolve, we are interested in exploring more efficient ways to integrate natural language comprehension into policy control, potentially reducing inference time and enhancing overall performance in future work.

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

# A  Training Architecture of Multi-style Policies

## A.1  Training Scenarios

To evaluate the effectiveness of the proposed LCMSP method, we utilized two open-source RL environments: the Highway environment[4] and the GRF environment[5]. Specifically, we employed the `highway-v0` task from the Highway environment. In the GRF environment, we designed single-player, two-player, and 5v5 scenarios to evaluate various multi-style policies and instructions. In total, four training scenarios were used in this study, as illustrated in Figure 7.

**Highway:** The Highway environment involves controlling a vehicle on a multilane highway populated with other vehicles. The agent's objective is to maintain a target speed while avoiding collisions with neighboring vehicles. We use the `highway-v0` task, which is configured with four lanes and populated with 50 vehicles; all other settings adhere to their default values. Both the frames per second (FPS) and the policy's inference frequency are set to 10 in this scenario. Each episode lasts for 40 seconds, equivalent to 400 steps. The episode is terminated prematurely if a collision occurs.

**Single-player and two-player scenarios in GRF:** These scenarios were designed to assess the capability of the proposed method to execute low-level instructions. Each episode is capped at a maximum of 400 steps and concludes once a goal is scored. To ensure diversity in training and policy generation, ball positions are randomly assigned across the field. One player's position is generated in close proximity to the ball, while the positions of other players are also randomly assigned on the field, except for goalkeepers, who remain within their box areas. In these scenarios, we control one team, while the opposing side is uncontrolled.

**5v5 scenario in GRF:** The 5v5 scenario is utilized to test the method's ability to execute complex high-level instructions. This scenario simulates a full 5v5 football game environment with two sides(Song et al., 2024), including various rules such as offside, out-of-bounds, corner kicks, red

---

[4]`https://highway-env.farama.org/installation/`
[5]`https://github.com/google-research/football`

cards, etc. Both sides are controlled by the trained policies. Each side manages four players in a five-player team, excluding the goalkeeper, who is controlled by the built-in AI. Each episode consists of 3,000 steps, with teams maintaining their respective sides throughout the game. At the beginning of each episode, the teams are positioned in a fixed formation, and the left/right sides are assigned randomly. The team scoring the most goals is declared the winner, and the game is considered a draw if two sides has same goals.

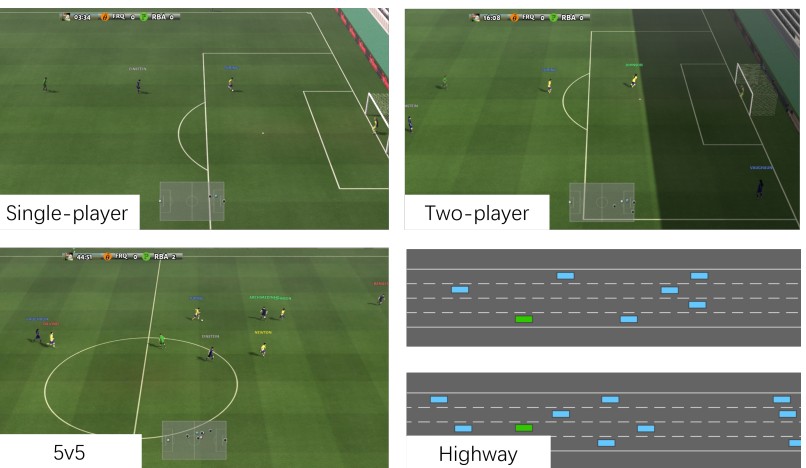

Figure 7: Training scenarios.

## A.2 FEATURES AND ACTION SPACES

**Highway:** In the Highway environment, we utilize the default `Kinematic` observation as the network input. The `Kinematic` observation is represented as a $V \times F$ array, where $V$ denotes the number of nearby vehicles that can be observed, and $F$ represents the number of features per vehicle. In this study, $V$ is set to eight, representing the eight nearest vehicles. Each vehicle is described by $F$ features, including attributes such as position, velocity, heading in radians, trigonometric heading, among others. These features are normalized and flattened into a one-dimensional vector to serve as the network input. We utilize the `Discrete Meta-Action` action set as the output, which comprises five discrete meta-actions: *LANE LEFT*, *IDLE*, *LANE RIGHT*, *FASTER*, and *SLOWER*, to control the vehicle.

**GRF:** Table 4 outlines the features designed for model training in the single-player, two-player, and 5v5 scenarios within the GRF environment, along with their respective lengths. The feature set for the *controlling player* includes attributes such as position, direction, speed, role, fatigue factor, areas, among others. The *ball state* includes the ball's position, direction, speed, distance to the controlling player, ownership, and other relevant attributes. The features for *teammates* and *opponents* are similar to those of the controlling player, with the addition of a one-hot player role indicator. The *nearest teammate* and *nearest opponent* features refer to the teammate and opponent closest to the controlling player. The *available actions* feature is a multi-hot indicator, where 1/0 signifies whether the corresponding action is available or unavailable to the controlling player in the current state. The *match state* logs information such as remaining time, scores, game mode, and so on. The *offside judgment* feature indicates if players are in an offside position. The *yellow/red cards* feature denotes whether players have received a yellow or red card. The *sticky action* feature indicates whether players are in a specific state, such as dribbling or sprinting. The *player active area* feature is a one-hot encoding that indicates the presence of players in each of the three horizontal and vertical zones of the field. The *team relative distance* feature calculates the average relative distance between the teams. The lengths of these features vary across modes due to the differing number of players. For the action space, we use the default action set of the GRF environment, comprising 19 actions.

Table 4: The content and length of the vector features for different scenarios in the GRF environment.

| Feature content | Single-player | Two-player | 5v5 mode |
|---|---|---|---|
| Controlling player | 19 | 19 | 19 |
| Ball state | 18 | 18 | 18 |
| Teammates | 9 | 18 | 36 |
| Nearest teammate | 9 | 9 | 9 |
| Opponents | 18 | 27 | 45 |
| Nearest opponent | 9 | 9 | 9 |
| Available actions | 19 | 19 | 19 |
| Match state | 10 | 10 | 10 |
| Offside judgment | 4 | 6 | 10 |
| Yellow / Red cards | 8 | 12 | 20 |
| Sticky action | 10 | 10 | 10 |
| Player distance to ball | 3 | 5 | 9 |
| Player active area | 6 | 6 | 6 |
| Team relative distance | 0 | 1 | 1 |
| Total | 142 | 169 | 221 |

### A.3   DESIGN OF META-BEHAVIOURS AND REWARD SHAPING

Table 5 provides a comprehensive overview of the meta-behaviors and their reward shaping for all training scenarios, which include the Highway environment and the single-player, two-player, and 5v5 scenarios within the GRF. The reward types are categorized into three kinds: *Basic*, *Bool*, and *Float*. The *Basic* reward is a constant that remains unaffected by changes in style parameters. The *Bool* and *Float* rewards correspond to the *Bool* and *Float* style parameters, respectively. The *Bool* type reward is binary, with 0 or 1 representing the deactivation or activation of the corresponding style preference. The *Float* type reward, on the other hand, can take on continuous values within a specified range, allowing for fine-grained control over the behavior.

**Highway:** The primary meta-behaviors in the Highway environment are *Speed*, *Time to Collision (TTC)*, *Lane Change*, and *Lane Preference*, which control the vehicle's behavioral style. We design four rewards corresponding to these meta-behaviors, which can be adjusted via style parameters. For example, the style parameter for the *Speed* reward ranges between [0, 1], encouraging the vehicle to maintain speeds between the minimum and maximum velocities accordingly. The *Lane (1, 2, 3)* style parameters represent preferences for the left lane, the two middle lanes, and the right lane, respectively. The *Collision* penalty is a *Basic* reward given to agents that collide with other vehicles.

**GRF:** In the GRF environment, the meta-behaviors include *Goal*, *Lose Goal*, *Get Possession*, and others. These rewards are distributed to all team members, regardless of which individual player completed the task. For instance, the *Shot* reward is allocated to all teammates, not just the player who actually took the shot, to foster teamwork among players. In the context of the football field:

- *Area X (1, 2, 3)* corresponds to the *Front*, *Midfield*, and *Back* areas, respectively.
- *Area Y (1, 2, 3)* denotes the *Left Wing*, *Central*, and *Right Wing* areas, respectively.
- *Shot Type (1, 2)* represents the *Goal Area Shot* and *Penalty Area Shot* actions.
- *Move Type (1, 2)* corresponds to *Run* and *Dribble* behaviors.
- *Formation Type (1, 2, 3)* indicates the *Retreat*, *Balanced*, and *Press* formations.
- *Spacing Type (1, 2, 3)* represents *Compact*, *Normal*, and *Loose* formations.

In the single-player and two-player modes, we categorize *Goal*, *Lose Goal*, and *Get Possession* as *Basic* rewards, primarily to evaluate the controllability of other meta-behaviors. In the 5v5 scenario, all meta-behaviors, including the *Win* preference, can be controlled and tested using style parameters.

### A.4   TRAINING ALGORITHM OF MULTI-STYLE RL POLICY

To efficiently train models with multi-style policies, we utilize a dual-clipping version (Ye et al., 2020) of Proximal Policy Optimization (PPO) (Schulman et al., 2017). PPO is a popular policy gra-

Table 5: Overview of meta-behaviors and their reward shaping for all training scenarios.

| Meta-Behaviors | Type | Range | Reward Shaping Content |
|---|---|---|---|
| Highway | | | |
| Collision | Basic | -1 | Penalize the vehicle for collisions |
| Speed | Float | [0, 1] | Encourage the vehicle to maintain a certain speed |
| Time to Collision | Float | [0, 1] | Encourage maintaining a distance from the vehicle ahead |
| Change Lane | Float | [0, 1] | Preference of lane changing during driving |
| Lane (1, 2, 3) | Bool | 0/1 | Encourage the vehicle to keep in the target lane |
| Single-Player Scenario in GRF Environment | | | |
| Goal | Basic | 0.1 | Encourage agents to score |
| Lose Goal | Basic | -1 | Penalize agents for conceding goals |
| Get Possession | Basic | 1 | Encourage agents to gain possession |
| Area X (1, 2, 3) | Bool | 0/1 | Encourage agents to arrive at the target Area X |
| Area Y (1, 2, 3) | Bool | 0/1 | Encourage agents to arrive at the target Area Y |
| Shot Type (1, 2) | Bool | 0/1 | Encourage agents to shot with the target type |
| Move Type (1, 2) | Bool | 0/1 | Encourage agents to move with the target type |
| Two-Player Scenario in GRF Environment | | | |
| Goal | Basic | 0.1 | Encourage agents to score |
| Lose Goal | Basic | -1 | Penalize agents for conceding goals |
| Get Possession | Basic | 1 | Encourage agents to gain possession |
| Hold Ball Preference | Bool | 0/1 | Preference of holding ball |
| Pass Preference | Bool | 0/1 | Preference of making effective passes |
| Formation Type (1, 2, 3) | Bool | 0/1 | Encourage the team to adopt the target formation |
| Shot Type (1, 2) | Bool | 0/1 | Encourage agents to shot with the target type |
| Move Type (1, 2) | Bool | 0/1 | Encourage agents to move with the target type |
| 5v5 Scenario in GRF Environment | | | |
| Win | Float | [0, 1] | Preference of winning |
| Goal | Float | [0, 1] | Preference of scoring goals |
| Lose Goal | Float | [0, 1] | Preference of preventing conceding goals |
| Hold Ball | Float | [0, 1] | Preference of holding ball |
| Get Possession | Float | [0, 1] | Preference of having possession |
| Formation | Float | [0, 1] | Encourage the team to adopt the target formation |
| Spacing | Float | [0, 1] | Encourage a target spacing between agents |
| Pass | Float | [0, 1] | Preference of making effective passes |
| Shot Type (1, 2) | Bool | 0/1 | Encourage agents to shot with the target type |
| Move Type (1, 2, 3) | Bool | 0/1 | Encourage agents to move with the target type |

dient algorithm that has demonstrated excellent performance in various tasks (Berner et al., 2019; Ouyang et al., 2022). It is designed to improve the stability and reliability of policy gradient methods, which directly optimize the policy to train agents. PPO limits changes in the policy by clipping the probability ratio between the current and old policies. The importance sampling ratio in PPO is defined as $r_t = \frac{\pi(a_t|s_t)}{\pi_{old}(a_t|s_t)}$, where $s_t$ and $a_t$ represent the state and action at time step $t$, respectively. Here, $\pi(a_t|s_t)$ and $\pi_{old}(a_t|s_t)$ are the probabilities of taking action $a_t$ in state $s_t$ under the current and old policies, respectively. We define the clipped ratio as $r_t^c = clip(r_t, 1-\varepsilon, 1+\varepsilon)$, where $\varepsilon$ is a hyperparameter. This clipping prevents large policy updates that could destabilize the training process. The policy objective is then defined as:

$$L_p = \begin{cases} -\hat{\mathbb{E}}_t[max(min(r_t\hat{A}_t, r_t^c\hat{A}_t)), \eta\hat{A}_t] & \hat{A}_t < 0 \\ -\hat{\mathbb{E}}_t[min(r_t\hat{A}_t, r_t^c\hat{A}_t)] & \hat{A}_t \geq 0 \end{cases} \tag{4}$$

where $\hat{\mathbb{E}}[...]$ denotes the empirical expectation over a finite batch of samples, and $A_t$ is the estimated advantage at time step $t$, computed via Generalized Advantage Estimation (GAE) (Schulman et al., 2016). The hyperparameters $\varepsilon$ and $\eta$ are the clipping parameters of the original PPO and the dual-clipped PPO, respectively. The value function objective in PPO is defined as:

$$L_v = \hat{\mathbb{E}}_t[(V(s_t) - G_t)^2] \tag{5}$$

where $G_t = V_{old}(s_t) + \hat{A}_t$ is the target return, and $V_{old}(s_t)$ is the value estimate from the old value function.

## A.5 NETWORK STRUCTURES

Figure 8 illustrates the network structures employed for model training in both the Highway and GRF environments. In the Highway environment, the state features and style parameters are initially processed through separate series of fully connected (FC) layers. The outputs of these layers are then concatenated to form a combined feature vector, which is shared by both the policy and value networks, as depicted in Figure 8(a). This concatenated tensor is subsequently fed into the policy network and the value network to generate their respective outputs. The network architecture for the GRF environment follows a similar design, as shown in Figure 8(b).

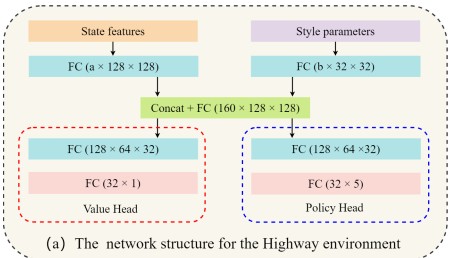 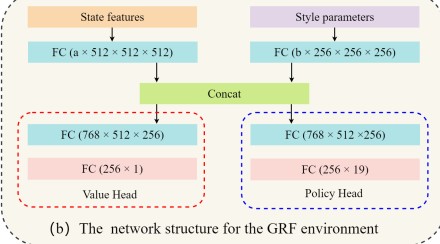

(a) The network structure for the Highway environment     (b) The network structure for the GRF environment

Figure 8: Network architectures for training models in the Highway and GRF environments. The dimensions labeled "a" and "b" in the blocks represent the sizes of the state features and style parameters, respectively. Leaky-ReLU is used as the activation function for the FC layers, except for the final layers. A Softmax layer is applied to the outputs of the policy head to convert them into action probabilities, with the final action selected by sampling based on these probabilities.

## A.6 TRAINING PROCESS

Throughout the training process across all scenarios, we maintained consistent hyperparameters. Table 6 lists the training hyperparameters utilized in our study. It is important to note that, given the vast number of possible hyperparameter combinations, we cannot guarantee that the selected hyperparameter values are optimal. However, we can confirm that agents trained with these hyperparameters demonstrate superior multi-style performance in both the Highway and GRF environments. The pseudocode of the training process is shown in Algorithm 1.

Table 6: The training hyperparameters.

| Hyperparameters | GRF | Highway |
|---|---|---|
| Batch size | 60,000 | 15,000 |
| Trajectory length | 128 | 64 |
| Sample reuse | About 1.0 | About 1.0 |
| PPO clipping | 0.2 | 0.2 |
| PPO dual-clipping | 3 | 3 |
| Gradient clipping | 25 | 30 |
| Discount factor $\gamma$ | 0.999 | 0.995 |
| GAE discount $\lambda$ | 0.95 | 0.95 |
| Value loss weight | 0.5 | 0.5 |
| Entropy coefficient | 0.02 | 0.05 |
| Optimizer | Adam | Adam |
| Learning rate | 5e-5 | 1e-4 |
| Adam $\beta_1, \beta_2$ | 0.99, 0.999 | 0.99, 0.999 |

Figure 9 presents the return curves for the Highway, single-player, and two-player scenarios, showing a steady increase as training progresses. Additionally, Figure 10 depicts the reward curves for each meta-behavior within the single-player and two-player scenarios in GRF. In the 5v5 scenario, rewards do not fully capture the model's effectiveness due to the involvement of self-play competition during training. Therefore, the effectiveness of the training in the 5v5 scenario can be assessed using the metrics provided in Appendix D.4. The training process utilized two NVIDIA A10 GPUs and 1,000 CPU pods. Each pod was allocated one CPU core and 2 GB of memory and consisted of the game client, one agent, and the models used by the agent. The training was implemented using

---

**Algorithm 1:** Training process of multi-style policy in LCMSP

---

**for** *each training epoch* **do**

    **for** *each episode* **do**

        Randomly generate a set of style parameter $\omega$;

        **for** *each environment step* **do**

            Get the states $s_t$, and execute actions $a_t$ by $\pi_\theta(a_t \mid s_t, \omega)$;

            Get the rewards and next states $r_t^\omega$ and $s_{t+1}$;

            Save $(s_t, \omega, a_t, r_t^\omega, s_{t+1})$ to the replay buffer;

    **for** *each gradient step* **do**

        Perform gradient step on $\theta$ by maximizing the expectation $J_{\pi_\theta}$;

        Perform gradient step on $\phi$ by minimizing the expectation $J_{V_\phi^\pi}$ *(if needed)*;

---

Python 3.8 with PyTorch 2.0. Unless otherwise specified, all results reported in this study are the averages over three replicated tests, each with different random seeds.

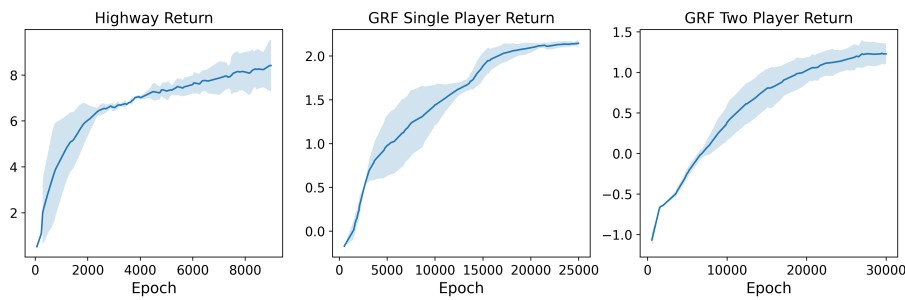

Figure 9: Training curves for Highway, single-player and two-player scenarios.

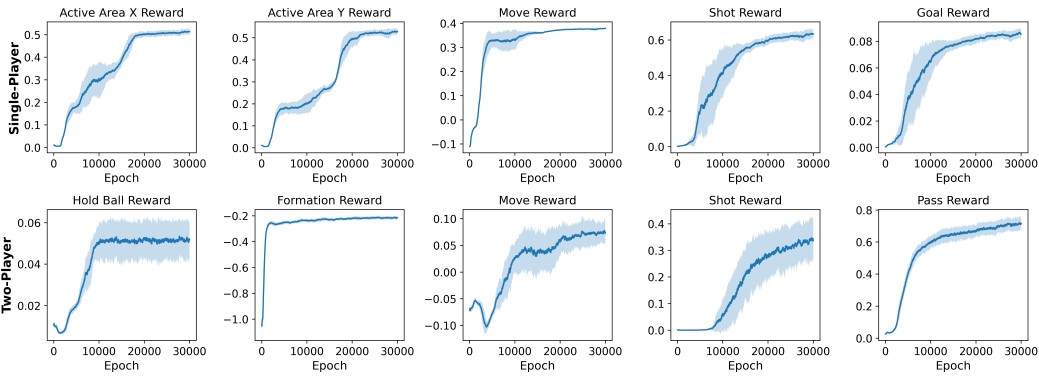

Figure 10: Training curves of meta-behaviors' rewards for single-player and two-player scenarios.

## B EXAMPLE OF DEGREE-TO-PARAMETER PROMPT

Table 7 presents an example prompt of the Degree-to-Parameter (DTP) method within the GRF 5v5 scenario. An instruction from the *Park the Bus* tactical set was selected to demonstrate how GPT-4o maps this instruction to style parameters.

Table 7: The prompt example for DTP method in GRF 5v5 scenario

**LLM Prompt**

In a football match, each team consists of five players: one goalkeeper and four outfield players. The match is played on a standard-sized field according to standard rules for 90 minutes. The goalkeeper and outfield players are controlled by different policies. Various styles govern the outfield players, operating simultaneously and interacting with each other. Each style can be adjusted on a scale from 0 to 10. Below are the explanations for each style:

Style-1: Focus on winning; higher values indicate a stronger desire to win. A value of 0 means no concern for winning.

Style-2: Focus on scoring goals; higher values indicate a stronger desire to score. A value of 0 means no concern for scoring.

Style-3: Focus on not conceding goals; higher values indicate a stronger desire to avoid conceding. A value of 0 means no concern for conceding goals.

Style-4: Focus on individual ball control; a value of 0 means decisions are situational, while a value of 10 means a preference for individual ball control.

Style-5: Focus on regaining possession; a value of 0 means no concern for possession, while a value of 10 means a strong emphasis on regaining possession.

Style-6: Focus on passing; a value of 0 means decisions are situational, while a value of 10 means a preference for passing.

Style-7: Relative distance between players; a value of 0 means compact spacing, while a value of 10 means more spread out.

Style-8: Overall team positioning; a value of 0 means closer to their own goal, while a value of 10 means closer to the opponent's goal.

The following styles are selected from specific options:

Style-9: Shooting distance; -1 means no preference, 0 means close range, and 1 means long range.

Style-10: Dribbling style; -1 means no preference, 0 means moderate speed, 1 means close control, and 2 means sprinting.

Based on an understanding of football and the styles above, carefully translate the instructions into corresponding style parameters, providing the reasoning process. Finally, provide a standard JSON format result where the keys are each style and the values are the corresponding style parameters.

Examples: **Insert some examples here for few-shot learning.**

Instruction: **Keep the defensive lines tight, sitting deep to prioritize shutting out the opposition's attack completely.**

**LLM Reponse**

To translate the instruction into corresponding style levels, let's break down the instructions and reason step by step:

1. Keep the defensive lines tight:

   This suggests maintaining a very compact and organized team shape.

   Style-7 (Relative distance between players): High importance for compactness, so a lower value around 2.

2. Sitting deep:

   This indicates positioning the team closer to their own goal.

   Style-8 (Overall positioning): High importance, so a very low value around 1.

3. Prioritize shutting out the opposition's attack completely:

   This suggests a very high emphasis on not conceding goals.

   Style-3 (Not conceding goals): Very high importance, so a maximum value of 10.

4. Other styles:

   Style-1 (Importance of winning): The focus is on defense rather than winning, so a lower value around 3.

   Style-2 (Scoring goals): No focus on scoring, so a low value around 2.

   Style-4 (Individual ball control): The instructions do not emphasize individual ball control, so a lower value around 3.

   Style-5 (Regaining possession): The instructions suggest a passive approach, so a moderate value around 4.

   Style-6 (Passing): Passing is not emphasized, so a moderate value around 4.

   Style-9 (Shooting distance): The instructions do not specify a preference for shooting distance, so a value of -1.

   Style-10 (Dribbling style): The instructions do not specify a dribbling style, so a value of -1.

Based on the reasoning above, the corresponding style levels are: 'Style-1': 3, 'Style-2': 2, 'Style-3': 10, 'Style-4': 3, 'Style-5': 4, 'Style-6': 4, 'Style-7': 2, 'Style-8': 1, 'Style-9': -1, 'Style-10': -1

## C  THE PROCESS OF INSTRUCTION GENERATION

To validate the capability of our proposed method and the comparison methods in following natural language instructions, and to train models for some of the comparison methods, we require a large set of instructions. Since we primarily test the ability to follow simple instructions in the Highway environment and two scenarios in the GRF environment, we leverage GPT-4o to generate reliable instruction sets.

The simple instructions are divided into five types: Normal, Short, Long, Unseen, and Inference. The generation methods for Normal, Short, and Long instructions are similar, with slight differences in the prompts used during generation. Taking the generation of Normal instructions in the single-player scenario as an example, the style parameters mainly control the agent's meta-behaviors towards *Area X*, *Area Y*, *Shot Type*, and *Move Type*. We firstly designed two sets of meta-behavior combinations: the navigation combination set and the shot combination set. The navigation combination set is composed of the meta-behaviors *Area X*, *Area Y*, and *Move Type*. By traversing all possible combinations of these three meta-behaviors, each combination can yield an instruction target. For each instruction target, the LLM generates 10 instructions that match its meaning. Therefore, in the Normal instruction type of the single-player scenario, the navigation combination set contains a total of 470 instructions for testing. The information of generated instructions is shown in Table 8. Similarly, the meta-behaviors in the shot combination set consist of *Shot Type* and *Move Type*. For each instruction target in all combinations, the LLM generates 10 instructions, totaling 80 instructions.

Table 8: The information of generated instructions.

| Scenario | Type | Set | Related meta-behaviours | Count |
|---|---|---|---|---|
| Highway | Normal, Short, Long | Set-1 | Speed, Change Lane | $50 \times 3$ |
| | | Set-2 | Speed, Lane preference | $100 \times 3$ |
| | | Set-3 | TTC, Lane preference | $70 \times 3$ |
| | Unseen | / | All | 90 |
| | Inference | / | All | 60 |
| Single-player | Normal, Short, Long | Navigate | Area X, Area Y, Move type | $470 \times 3$ |
| | | Shot | Shot type, Move type | $80 \times 3$ |
| | Unseen | / | All | 60 |
| | Inference | / | All | 50 |
| Two-player | Normal, Short, Long | Ball control | Formation type, Pass preference, Hold ball preference | $100 \times 3$ |
| | | Shot | Shot Type, Move type | $70 \times 3$ |
| | Unseen | / | All | 60 |
| | Inference | / | All | 50 |
| 5v5 | Abstract | / | All | 180 |

The prompts used for generating the Normal type instructions are shown in Table 9. The prompts for generating the Short and Long type instructions are similar, with minor modifications to adjust the instruction length. The Unseen type instructions are used to test the method's generalization ability to target instructions that have never been seen before. These Unseen instructions are generated by extracting certain instructions from the Normal, Short, and Long types to form a test set. For example, in the single-player scenario, the instructions of Normal, Short, and Long types generated by the style "*Area X*: 1, *Area Y*: 1, *Move Type*: 0" are extracted as part of the Unseen instruction set. Several style combinations are selected for extraction. When testing with Unseen instructions, similar examples used as few-shots in the prompt are removed, and the instructions corresponding to the target style are not used during model training in other comparison methods. The Inference type instructions are designed to test the method's inference ability, all Inference type instructions require logical deductions to determine the correct targets. The prompts used for generating Inference type instructions include extra content and examples to enable the LLMs to perform logical deduction.

Table 9: An example of prompt for generating shot combination set instructions in Normal instruction type.

You are an assistant for a football game, and you need to describe executable commands in the game using natural language. Below are explanations and examples of the commands:

Positions: Penalty Area, Goal Area

Movement actions include: Run, Dribble

Shooting actions include: Shoot

Commands: Commands are combinations of movement, position, and shooting.

Example 1

Command: "Run, Penalty Area, Shoot"

Natural language: 1. High-speed powerful long shot. 2. Sprint over and fire in the 18-yard Box.

Example 2

Command: "Dribble, Goal Area, Shoot"

Natural language: 1. Control the ball well, get as close to the goal as possible before shooting. 2. Dribble past the opponent first, then shoot within the 6-yard Box.

Example 3

Command: "Run, ' ', Shoot"

Natural language: 1. Move to a suitable position and shoot. 2. Run to the shooting position, shoot quickly.

Example 4

Command: "' ', Goal Area, Shoot"

Natural language: 1. The opponent's goalkeeper is very focused, long shots are not suitable, only point-blank shot has a chance. 2. There is no opponent in the 6-yard Box, tap-in the ball.

For each command, generate ten diverse natural language descriptions, separated by line breaks, starting with a numerical ID.The generated sentences must have different lengths, different punctuation, and different word orders. Use the aliases for Large Penalty Area, Small Penalty Area, and shooting.

Command: {command}

Natural language:

In the single-player and two-player scenarios, only *Bool* style parameters are used. Therefore, we can generate the corresponding instructions by combining the two states (Deactivated or Activated) of the related meta-behaviors in the combination set. The Highway environment includes both *Float* and *Bool* style parameters, and the *Float* style parameters can be described using 11 degrees in this study. To simplify the testing process for simple instructions, we test only the extreme states of the related meta-behaviors, corresponding to style parameters of zero or one. To test complex and abstract high-level instructions, we designed six tactics analogous to real-world football strategies in the 5v5 scenario of GRF. For each tactic, we provided three examples in the prompt and then used GPT-4o to generate 30 natural language instructions. The generated instructions require a certain level of expertise to understand and necessitate the coordination of multiple meta-behaviors to execute. Therefore, we consider these instructions to be complex and abstract high-level commands. Finally, we generated a total of 180 instructions that satisfy these criteria in the 5v5 scenario.

## D  ADDITIONAL EXPERIMENT RESULT

### D.1  SUCCESS CRITERIA FOR INSTRUCTION EXECUTION

In all scenarios other than the 5v5 scenario, the instructed behavior styles are combinations of multiple distinct meta-behaviors, as illustrated in Table 8. To automate the evaluation of whether each instruction is correctly executed, we designed corresponding evaluation criteria for each meta-behavior. An instruction is considered correctly executed if all the specified meta-behaviors meet their respective evaluation standards. For instance, for the instruction "*Dribble past the opponent first, then shoot within the 6-yard box*", the corresponding styles are "*Move Type*: *Dribble* and *Shot Area*: *Goal Area*". To determine correctness, we established the following criteria: the agent must perform *Dribble* actions for more than 80% of its ball-carrying movements; the final shooting action must occur within the *Goal Area*; and a goal must be scored.

Table 10 lists the success criteria for determining each meta-behavior in the Highway environment, as well as in the single-player and two-player scenarios in GRF. In the Highway environment, all criteria include the precondition that the vehicles do not collide during the episode. In the complex 5v5 scenario, we employ abstract high-level instructions that are difficult to evaluate for successful execution using rule-based methods. Therefore, we rely on statistical analysis to assess the performance of instruction-following ability in the 5v5 scenario, rather than using specific success criteria for instruction execution.

Table 10: Criteria of success execution for each meta-behavior.

| Meta-behaviour | Success criteria |
|---|---|
| Highway | |
| Speed | The average speed of the vehicle falls within a target speed range. |
| Time to Collision | The average TTC of the vehicle falls within a target range. |
| Lane Changing | The ratio of left and right lane-change actions falls within a target range. |
| Lane Preference (1, 2, 3) | The proportion of time spent in the target lane exceeds a certain threshold. |
| Single-Player Scenario in GRF | |
| Area X (1, 2, 3) | The player's average x-coordinate falls within the corresponding range. |
| Area Y (1, 2, 3) | The player's average y-coordinate falls within the corresponding range. |
| Shot Type (1, 2) | A goal is scored with the final shot taken within a designated area. |
| Movement Type (1, 2) | 80% of ball possession movements correspond to designated actions. |
| Two-Player Scenario in GRF | |
| Ball Possession Preference | 80% of the steps involve our player holding the ball. |
| Pass Preference | The number of successful passes is greater than 10. |
| Formation Type (1, 2, 3) | The team's average x-coordinate falls within the corresponding range. |
| Shot Type (1, 2) | A goal is scored with the final shot taken within a designated area. |
| Movement Type (1, 2) | 80% of ball possession movements correspond to designated actions. |

## D.2 HIGHWAY ENVIRONMENT

**Multi-style policy evaluations.** To assess the performance of the trained multi-style policies, we conducted a series of experiments to determine the success rate of executing the correct behavioral styles based on the provided multi-style parameters. To ensure consistency with the behavioral styles specified by the instructions, we evaluated policy performance using the combinations of meta-behaviors and style parameters from the generated instructions, as summarized in Table 8. For unspecified style parameters, *Float*-type parameters were set to 0.5, and *Bool*-type parameters were set to 0. For instance, in the Set-1 combination set of the Highway environment, the style "*Speed*: Fast, *Lane Change*: Frequent" has the style parameters for *Speed* and *Change Lane* set to 1, while the unspecified parameters for *TTC* (Time to Collision) and *Lane Preference* are set to 0.5 and 0, respectively.

Table 11 presents the success rates of executing the correct behavioral styles based on the provided style parameters. As shown, the multi-style policy demonstrates satisfactory performance in executing different combinations of meta-behaviors in the Highway environment. The Set-3 combination set exhibits a relatively lower success rate because the *TTC* is difficult to control perfectly due to the precondition of avoiding collisions with other vehicles.

Table 11: Success rates of executing the correct behavioral style with different combinations of multi-style parameters.

| Scenario | Combination Set | Related meta-behaviors | Success rate |
|---|---|---|---|
| Highway | Set-1 | Speed, Change Lane | 93.77% |
| | Set-2 | Speed, Lane preference | 93.28% |
| | Set-3 | TTC, Lane preference | 87.94% |
| Single-player | Navigate | Area X, Area Y, Move type | 90.43% |
| | Shot | Shot type, Move type | 96.50% |
| Two-player | Ball control | Formation type, Pass preference, Hold ball preference | 98.03% |
| | Shot | Shot Type, Move type | 90.04% |

### D.3 Single-player and two-player scenarios in GRF

**Multi-style policy evaluations.** Table 11 presents the success rates for executing the correct behavioral styles in the single-player and two-player scenarios in GRF. The results indicate that the trained multi-style policy achieves a high success rate for each behavioral style in both scenarios. The style parameters tested here are generated similarly to those used in the multi-style policy evaluations of the Highway environment.

**The implementation of TALAR method.** In our experiments, the TALAR method translates natural language instructions into style parameters to serve as inputs for RL policies. The translator is implemented by fine-tuning a BERT model. We utilized 80% of the Normal, Long, Short, and Inference type instructions as the training set. The Unseen instructions and the remaining portions of the other types were used as the testing set. In the single-player and two-player scenarios, the style parameters are all discrete *Bool* types. The translation process can be modeled as a multi-class classification task, where each instruction's label corresponds to the style parameters associated with each meta-behavior used during its generation.

During the training process of the translator, the BERT model's parameters are frozen, and instructions are input to obtain the pooler output. This output provides a fixed-size representation of the input sequence and is part of BERT's architecture learned during pre-training. We pass the pooler output through an additional FC layer with multiple output heads to achieve multi-class classification results, training only the parameters of this added layer.

**Alignment accuracy.** Figure 11 shows the alignment accuracy for each type of instruction in the single-player and two-player scenarios. It can be observed that the alignment effectiveness varies across different LLMs, but the differences are not substantial. The TALAR method generally underperforms compared to the LCMSP method, especially on Unseen type instructions whose behavioral styles were not encountered during training. The TALAR method also exhibits poor comprehension of long instructions. This is because the BERT model summarizes long sequences into a single [CLS] token for classification.

Comparing with Figure 4, we observe that alignment success rates can sometimes be lower than the execution success rates. This occurs because alignment success rates indicate that the generated style parameters precisely match the ground truth specified by the instructions. However, in practice, the policy may still execute the correct behavioral style even with incorrect style parameters. For instance, if the instruction is to dribble into the penalty area and shoot, but the alignment only includes shooting, the policy might randomly choose between dribbling or sprinting. This can result in alignment omissions or errors, yet there remains a chance of executing the instruction correctly.

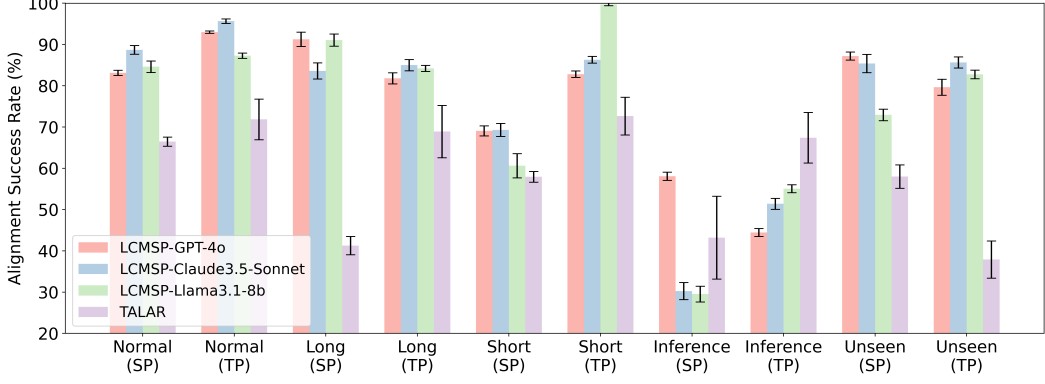

Figure 11: Alignment success rate on single-player and two-player scenarios. SP and TP represent single-player and two-player scenarios, respectively

### D.4 5V5 SCENARIO IN GRF

#### D.4.1 STYLE PARAMETER OF DIFFERENT TACTICS

In the 5v5 scenario, LCMSP derives the style parameters for the RL policy from high-level abstract instructions. To validate this process, we used GPT-4o to map each instruction to the corresponding style parameters ten times and plotted the results in Figure 12. It shows that the *Positive Attack* tactic has *Win*, *Goal*, and *Formation* parameters slightly above the neutral value (0.5). The *All-Out Attack* tactic exhibits higher values for *Win* and *Goal*, with less emphasis on *Lose Goal* prevention, and places greater emphasis on *Get Possession* and *Hold Ball*. The *Balanced Play* tactic has all parameters close to the neutral value of 0.5. The *Counter Attack* tactic maintains some inclination towards *Win* and *Goal* but places greater emphasis on *Lose Goal* prevention, featuring the deepest *Formation* to allow space for counterattacks. In contrast, the *Park the Bus* tactic exhibits very low tendencies for *Win* and *Goal*, as well as lower values for *Get Possession*, *Pass*, and *Hold Ball*, while placing an extremely high emphasis on *Lose Goal* prevention. The *Tiki-Taka* tactic shows the highest values for *Get Possession* and *Pass*. These results demonstrate that LCMSP can accurately align abstract natural language instructions with the corresponding style parameters and capture the magnitude of each style.

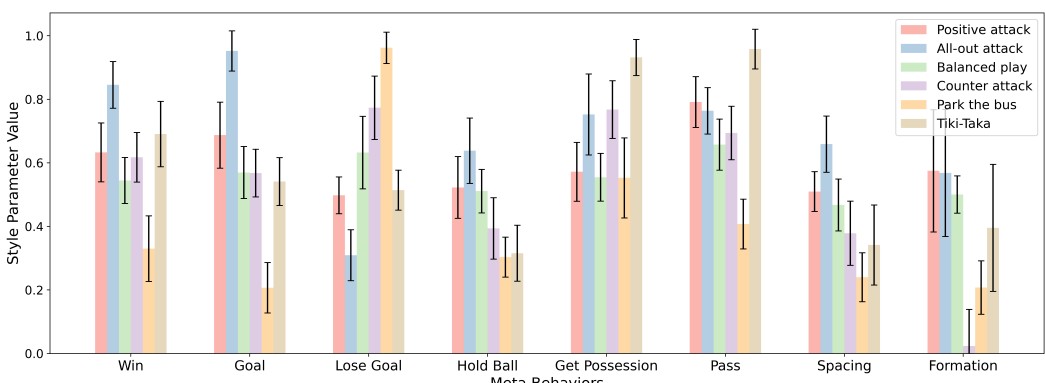

Figure 12: Style parameters under different tactical instructions.

#### D.4.2 FINE-GRAINED ADJUSTMENT OF STYLE PARAMETERS

To demonstrate that the styles trained by our method can be finely adjusted based on style parameters, we selected six *Float* type style parameters in the 5v5 scenario. For each style parameter, we fixed the other style parameters at their baseline values and sampled 20 values from 0 to 1 in increments of 0.05. For each sampled value, we ran 1,000 episodes against the same model with random styles. Figure 13 illustrates the corresponding in-game metrics as functions of the style parameters. It can be observed that, for each style parameter, adjusting its value results in a nearly linear and smooth change in the corresponding metric.

#### D.4.3 IN-GAME METRICS UNDER DIFFERENT POLICY STYLES

To demonstrate the effects of individual style parameters, we established a baseline where all *Float* type style parameters are set to 0.5, and all *Bool* type style parameters are set to 1 in this testing. For each style parameter under test, we set its value to the extreme of either 0 or 1 while keeping all other style parameters at their baseline values. We then ran 1,000 episodes for each testing style against the same model with random styles and calculated the mean of the corresponding in-game metrics, as presented in Table 12. It can be observed that for each testing style parameter, the associated metrics corresponding to parameter values of 0 and 1 reach their maximum and minimum values, respectively, except for the *Lose Goal* style parameter. Since the *Lose Goal* parameter represents the magnitude of the penalty for conceding a goal, it exhibits an inverse relationship with the number of goals conceded. When the *Lose Goal* parameter is set to 1, the number of goals conceded is minimized. Conversely, setting it to 0 leads to a significant increase in goals conceded, indicating

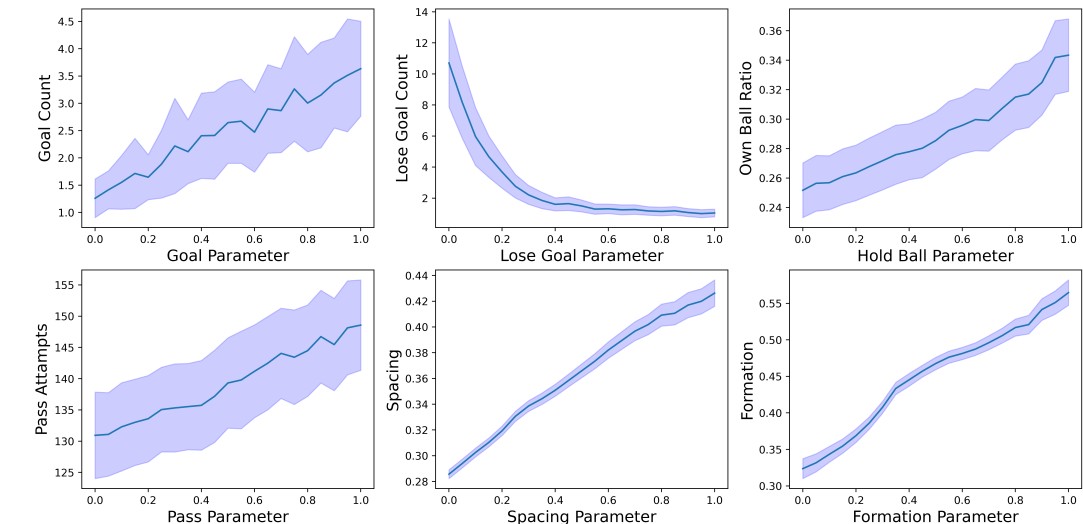

Figure 13: Fine-grained adjustment of style parameters and their corresponding changes in in-game metrics. Note that the *Lose Goal* parameter is inversely related to the number of goals conceded because its magnitude represents the penalty for conceding a goal. When the *Lose Goal* parameter becomes very low, the number of goals conceded increases significantly, indicating that conceding goals is scarcely penalized, leading the agent to disregard defense.

that conceding goals is scarcely penalized, and the agent disregards defensive play. Taking the *Goal* style parameter as an example, when set to 1, the number of goals scored is the highest among all styles. Conversely, when set to 0, the number of goals scored significantly decreases. For *Bool* type style parameters such as *Shot Type*, activating the *Shot (Goal Area)* style results in an average of 0.77 shots in the goal area, the highest among all styles. In contrast, when the *Shot (Penalty Area)* style is activated, the number of shots in the goal area decreases to 0.26.

There are also interactions between style parameters. For instance, when the *Spacing* style parameter is set to 0, the team's formation becomes very compact, resulting in the number of passes increasing to 163, even higher than when the *Pass* style parameter is activated, because passing becomes easier when players are closer together. Conversely, when the *Spacing* parameter is set to 1, the number of passes correspondingly decreases because players are too far apart, making passing more difficult. Similarly, setting the *Formation* style parameter to 1 causes the team to press too far forward, leading to a significant increase in the number of goals conceded and a corresponding increase in shots against in the goal area. Conversely, when the *Formation* parameter is set to 0, the team tends to stay in the defensive half, resulting in fewer goals conceded and a corresponding decrease in the number of ball possession turnovers.

## E  EXAMPLES OF EMPLOYED INSTRUCTIONS

In each scenario, we selected three instruction examples for each instruction type. The instruction examples for the Highway environment are presented in Table13. The instruction examples for the single-player and two-player scenarios are provided in Tables14 and 15, respectively. Similarly, for the 5v5 scenario in GRF, we selected three instructions for each tactic, as presented in Table 16.

Table 12: In-game metrics with different styles in 5v5 scenario.

| | Win rate | Score | Lost Score | Shot | Pass | Hold Ball | Goal Area Shot | Penalty Area Shot | Get Possession | Lost Possession | Spacing | Formation | Run | Dribble | Sprint |
|---|---|---|---|---|---|---|---|---|---|---|---|---|---|---|---|
| Win-0 | **0.53** | 2.82 | 1.35 | 28.15 | 133.4 | 0.29 | 0.32 | 27.82 | 20.19 | 19.82 | 0.37 | 0.47 | 0.23 | 0.72 | 0.05 |
| Win-1 | **0.57** | 2.4 | 1.53 | 26.3 | 137.91 | 0.3 | 0.3 | 26.01 | 20.61 | 20.39 | 0.36 | 0.47 | 0.21 | 0.74 | 0.05 |
| Goal-0 | 0.34 | **1.39** | 2.52 | 21.25 | 136.84 | 0.28 | 0.15 | 21.09 | 17.6 | 18.73 | 0.39 | 0.46 | 0.21 | 0.75 | 0.03 |
| Goal-1 | 0.59 | **3.55** | 1.28 | 30.72 | 122.78 | 0.3 | 0.49 | 30.22 | 21.24 | 20.33 | 0.36 | 0.49 | 0.28 | 0.66 | 0.06 |
| Lose Goal-0 | 0.22 | 1.87 | **10.91** | 23.23 | 132.26 | 0.23 | 0.26 | 22.97 | 14.04 | 17.81 | 0.35 | 0.51 | 0.21 | 0.75 | 0.03 |
| Lose Goal-1 | 0.54 | 2.72 | **1.02** | 24.91 | 124.93 | 0.32 | 0.32 | 24.59 | 20.12 | 19.51 | 0.36 | 0.46 | 0.24 | 0.71 | 0.05 |
| Hold Ball-0 | 0.48 | 2.79 | 1.99 | 29.98 | 146.7 | **0.25** | 0.36 | 29.62 | 20.87 | 20.91 | 0.36 | 0.47 | 0.21 | 0.74 | 0.04 |
| Hold Ball-1 | 0.47 | 2.02 | 1.06 | 22.71 | 124.37 | **0.34** | 0.31 | 22.41 | 18.87 | 18.69 | 0.37 | 0.46 | 0.25 | 0.7 | 0.05 |
| Get Possession-0 | 0.46 | 2.54 | 1.82 | 31.27 | 128.32 | 0.25 | 0.36 | 30.9 | **20.55** | **20.61** | 0.38 | 0.47 | 0.26 | 0.69 | 0.04 |
| Get Possession-1 | 0.49 | 2.66 | 1.18 | 24.43 | 137.52 | 0.32 | 0.29 | 24.14 | **19.99** | **19.51** | 0.36 | 0.47 | 0.23 | 0.71 | 0.06 |
| Pass-0 | 0.5 | 2.56 | 1.49 | 27.95 | **130.9** | 0.29 | 0.34 | 27.61 | 20.42 | 20.23 | 0.37 | 0.47 | 0.22 | 0.74 | 0.04 |
| Pass-1 | 0.47 | 2.39 | 1.41 | 26.3 | **149.75** | 0.28 | 0.32 | 25.97 | 20.73 | 20.57 | 0.36 | 0.47 | 0.25 | 0.69 | 0.06 |
| Spacing-0 | 0.5 | 2.48 | 1.49 | 29.24 | 163.29 | 0.29 | 0.35 | 28.89 | 22.01 | 21.91 | **0.29** | 0.49 | 0.22 | 0.73 | 0.05 |
| Spacing-1 | 0.33 | 2.62 | 3.89 | 26.3 | 118.43 | 0.24 | 0.45 | 25.85 | 17.79 | 19.12 | **0.43** | 0.49 | 0.25 | 0.7 | 0.05 |
| Shot (Goal Area) | 0.49 | 2.73 | 1.52 | 29.9 | 136.89 | 0.29 | 0.77 | 29.13 | 20.86 | 20.55 | 0.36 | 0.48 | 0.27 | 0.66 | 0.07 |
| Shot (Penalty Area) | 0.49 | 2.54 | 1.58 | 27.38 | 139.7 | 0.28 | 0.26 | 27.12 | 20.4 | 20.24 | 0.37 | 0.47 | 0.23 | 0.72 | 0.05 |
| Move (Run) | 0.52 | 2.76 | 1.7 | 29.08 | 144.12 | 0.27 | 0.37 | 28.71 | 21.07 | 20.83 | 0.37 | 0.47 | **0.45** | 0.52 | 0.03 |
| Move (Dribble) | 0.49 | 2.5 | 1.81 | 27.71 | 146.9 | 0.28 | 0.29 | 27.42 | 20.35 | 20.35 | 0.37 | 0.47 | 0.12 | **0.87** | 0.01 |
| Move (Sprint) | 0.48 | 2.69 | 1.87 | 28.19 | 146.69 | 0.26 | 0.32 | 27.86 | 20.79 | 20.7 | 0.36 | 0.47 | 0.22 | 0.57 | **0.21** |
| Formation-0 | 0.28 | 1.26 | 1.14 | 15.62 | 114.62 | 0.3 | 0.18 | 15.44 | 13.84 | 14.28 | 0.41 | **0.32** | 0.26 | 0.67 | 0.07 |
| Formation-1 | 0.31 | 2.9 | 8.51 | 26.64 | 116.08 | 0.23 | 0.65 | 25.99 | 15.58 | 18.16 | 0.33 | **0.56** | 0.25 | 0.71 | 0.04 |

Table 13: Examples instructions for Highway environment

| Instruction Type | Instruction Examples |
|---|---|
| Normal | Keep a larger distance on the left lane while driving. |
| | Increase your speed and change lanes as often as possible on the road. |
| | Push the pedal to the metal and steer your way into the fast lane. |
| Long | Increase the vehicle's following space by moving into the left lane, ensuring a safer gap to the car ahead, thus providing ample room for unexpected maneuvers while driving. |
| | Speed up and make frequent lane changes to optimize your travel time and maneuver through traffic more efficiently, ensuring you're adeptly navigating the road to reach destinations quickly. |
| | There's an open path in the fast lane. Quickly hit the accelerator and make the switch to ensure a smooth journey at a faster speed. |
| Short | Shift left, keep a larger space. |
| | Quickly drive and switch lanes often. |
| | Increase speed in the fast lane. |
| Unseen | Take slow lane, widen the space. |
| | Maintain a wider space while driving in the slow lane. |
| | Switch to the slow lane while maintaining a more substantial distance from the car ahead, prioritizing safety by allowing more reaction time and space for sudden maneuvers. |
| Inference | There are a lot of vehicles now, but there are still gaps. I need to go there quickly. |
| | I'm feeling uneasy about the speed of traffic in the left lane, so I'll just stick to the calmer side here. |
| | The open road stretches ahead invitingly, unhindered by other vehicles. |

Table 14: Examples instructions for for single-player scenario

| Instruction Type | Instruction Examples |
|---|---|
| Normal | Shift the play to the left front with controlled dribbling. |
| | Fill the midfield gap to create scoring opportunities. |
| | Rush to the right-back area to bolster our defense |
| Long | Ensure you dribble the ball towards the left front, as it will help you position yourself for an aggressive push towards the opponent's goal while evading their central defenders. |
| | A well-positioned midfielder is key to controlling the game; ensure your movements are calculated to intercept passes and assist in both defensive and offensive maneuvers. |
| | Quickly fall back to the right side of the back pitch to bolster our defensive stance against their attacking players and protect our goal from imminent threats. |
| Short | Dribble to the front left. |
| | Control the center |
| | Hurry to the right rear. |
| Unseen | Dribble to the Goal Box, hit it. |
| | Get past opponents, get into the Goal Box, and shoot for the goal. |
| | Control the ball skillfully, maneuver past defenders, and position yourself optimally for a shot within the 6-yard Box. |
| Inference | Our midfield has been breached, and the opponent's attack this time poses a significant threat. |
| | The opposing team's defenders are clustered in the center, leaving a clear lane down the left flank for a potential breakthrough. |
| | A gap has opened up in the penalty box, and it appears the goalkeeper is vulnerable, offering a perfect chance to score. |

Table 15: Examples instructions for two-player scenario

| Instruction Type | Instruction Examples |
|---|---|
| Normal | The team pulls back as one player keeps the ball at their feet. |
| | While evenly distributed, execute precise passes across the field. |
| | Organize a high press, pushing all players to advance cohesively. |
| Long | To mitigate the risk of losing possession amid such aggressive opposition, our immediate action should be to retake our positions in the backcourt while honing our dribbling capabilities to respond effectively to their pressure. |
| | Maintain equilibrium in both offensive and defensive lines; utilize frequent passing to pull the opponent's formation apart while searching for gaps and opportunities to exploit their weaknesses. |
| | Encourage a high-press system where everyone embodies the spirit of teamwork; as the ball is moved forward, even the non-ball holders must be vigilant, closing down on the rival's players to eliminate their options. |
| Short | Step back while holding onto the ball. |
| | Maintain balance and pass the ball effectively. |
| | Move forward together. |
| Unseen | Sprint and hit |
| | Sprint into position and fire at the goal |
| | Leverage your agility to break through the defense line, finding an opportune moment to take a shot immediately. |
| Inference | The match is reaching a critical phase, and we must focus on defending to secure our lead. |
| | The opponent's defense is currently well-organized, and we need to find a way to create scoring opportunities through teamwork. |
| | As we accelerate towards the heart of their defensive setup, the opportunity for a crucial finish is within reach. |

Table 16: Examples instructions for 5v5 scenario

| Tactics | Instruction Examples |
|---|---|
| Positive attack | Utilize precise forward momentum, balancing aggression with thoughtful possession. |
| | Keep control and intent in attacking plays, using strategic movements to create impactful opportunities. |
| | Apply steady pressure with calculated advances, optimizing space and timing to open defenses. |
| All-out attack | Push the entire squad forward, embrace an aggressive mindset, and prioritize scoring over defense. |
| | Move all players upfield, commit to aggressive attacking, and maintain high pressure on their backline. |
| | Focus entirely on creating goal threats, push the whole team offensively, and allow defensive gaps as needed. |
| Balanced play | Maintain equilibrium on the field by harmonizing defensive duties with attacking opportunities. |
| | Balance offensive creativity with defensive discipline to secure control over the game. |
| | Keep the lines compact and organized, supporting both defenders and attackers equally. |
| Counter attack | Keep a low block, prioritize defensive duties, and break forward with purpose when opportunities arise. |
| | Allow the opposition to commit forward, then initiate swift counter movements with few, precise passes. |
| | Hold a compact shape, absorb pressure, and strike swiftly with direct counter-attacks when the ball is won back. |
| Park the bus | Adopt an ultra-defensive posture, minimizing offensive efforts to prioritize preserving our clean sheet. |
| | Take a no-risk approach, filling the pitch with defensive bodies to crowd out any offensive threats. |
| | Close down all spaces at the back, maintaining a sturdy defensive shape to prevent any breakthrough from the opposition. |
| Tiki-Taka | Execute comprehensive pressing strategies to recover possession quickly, then maintain control using Tiki-Taka principles. |
| | Position for high intensity pressing meant to stymie attacks, paired with fast-paced, controlled ball movement. |
| | Maximize on high pressing to regain control swiftly, leveraging it through continuous short-passing plays. |

