# OpenReview forum: "Language-conditioned Multi-Style Policies with Reinforcement Learning"
_ICLR.cc/2025/Conference — Submitted to ICLR 2025_

### Official Review · Reviewer_eemt · 2024-10-21

**Soundness:** 3
**Presentation:** 3
**Contribution:** 2
**Rating:** 5
**Confidence:** 2

**Summary:**

The paper introduces LCMSP, a method that enables RL agents to follow abstract language instructions and adopt diverse behavioral styles using large language models (LLMs). It achieves this by aligning instructions with style parameters, allowing agents to handle complex environments and tasks effectively.

**Strengths:**

- The proposed method, LCMSP, integrates reinforcement learning with high-level language instructions through the use of style parameters, allowing for flexible control of agent behavior.
- The evaluation is thorough, covering different environments like autonomous driving and football simulations, which demonstrates the method’s ability to handle diverse and complex tasks.
- Results indicate high alignment accuracy and execution success rates, proving the approach's effectiveness in following abstract instructions.

**Weaknesses:**

- The method is ad-hoc. Designing meta-behaviors and style parameters is currently manual and requires careful consideration. Providing guidelines or automated approaches for defining these components could enhance usability and scalability in new domains.

- The evaluation mainly covers common instruction types, but extending tests to include more ambiguous or cross-lingual instructions would strengthen claims about generalizability and adaptability to diverse scenarios.

- The paper could benefit from more comparisons with other LLM-based RL methods to better demonstrate the effectiveness and uniqueness of its approach, specifically in terms of instruction alignment.

**Questions:**

- Is there a specific strategy or set of guidelines you recommend for designing meta-behaviors and style parameters in new environments? Are there automated or semi-automated methods that could aid in this process, especially when scaling to new and diverse domains?

- Could you provide additional tests or discussions on the model’s ability to generalize to ambiguous or cross-lingual instructions? It would help to understand if the LCMSP framework can effectively adapt to more varied linguistic inputs and thereby improve claims about its broad applicability.

- How does LCMSP compare with other LLM-based RL approaches in terms of aligning language instructions with policy behaviors? It would be helpful to see direct comparisons or analysis, particularly focused on alignment efficiency and adaptability, to better evaluate the contribution's uniqueness and robustness.

---

> ### Author Response · Authors · 2024-11-20
> **Response to Weaknesses 1-2**
>
> **W1**：Thank you for your feedback. We acknowledge that designing meta-behaviors is currently a heuristic process that requires a certain level of understanding of the environment or task. During our exploration in this study, we have developed two design principles to guide this process:
>
> ●**Minimal Meta-Behaviors**: Use as few meta-behaviors as possible to cover the widest range of agent behaviors within the environment.
>
> ●**Facilitate Reward Shaping**: Ensure that meta-behaviors are designed to facilitate easy reward shaping.
>
> This guidance is akin to the "Occam's Razor" principle, where we aim to describe the majority of an AI agent's behaviors using fewer and simpler elements. This can be achieved through various combinations and degrees of these elements, aligning with the concept of multi-style policy training based on meta-behaviors.
>
> In line with these principles, researchers can design corresponding meta-behaviors based on the desired policy styles for different environments. Currently, there is no totally automated method for designing meta-behaviors, much like reward shaping process in RL, which requires designing different rewards for different environments. It can be likened to "labeling" in supervised learning and is essential for the AI research. Fortunately, recent studies have begun exploring the use of LLMs for automated reward shaping. Following the reviewer's suggestion, we believe leveraging LLMs to design meta-behaviors is a promising research direction, which can increase the automation of our method.
>
> Additionally, typical language-conditioned RL typically requires designing rewards for behaviors corresponding to each instruction. For complex instructions like "Counter Attack," it is challenging to design rewards that assess how well an agent performs this instruction. However, through the combination of meta-behaviors, such complex behaviors can be trained. Decomposing complex instructions into meta-behaviors inherently reduces the burden of designing rewards. This approach allows for the emergence of sophisticated behaviors without the need to explicitly define rewards for each complex instruction.
>
> **W2**: We sincerely appreciate the reviewer's insightful suggestion. In response, we have conducted two new experiments to evaluate our method's performance on ambiguous instructions and cross-lingual instructions.
>
> For the ambiguous instructions experiment, we designed a protocol to simulate typographical errors that commonly occur during human input. Specifically, for each character in the instruction, we randomly replaced it with an adjacent character on the keyboard with a 3% probability like in [1][2]. This approach mimics realistic scenarios where users may make typing mistakes.
>
> To create the cross-lingual instructions dataset, we uniformly translated the original instructions into Spanish, Chinese, or French, while maintaining some in English. This diverse linguistic mix allows us to test our method's robustness across multiple languages.
>
> Both of these modified instruction sets were based on the GRF single-player instruction set. The results of our method's are presented in Table R1:
>
> Table R1: Alignment accucary of LCMSP-GPT4o on ambiguous and cross-lingual instruction sets with GRF single-player scenario.
>
> | Instruction Type | Original     | Ambiguous    | Cross-lingual |
> |------------------|--------------|--------------|---------------|
> | Normal           | 83.13±0.62%  | 80.17±0.85%  | 79.50±0.71%   |
> | Long             | 91.27±1.73%  | 86.67±3.70%  | 84.17±2.72%   |
> | Short            | 69.07±1.23%  | 67.50±2.16%  | 67.50±0.71%   |
> | Inference        | 58.07±1.00%  | 54.17±2.25%  | 50.83±1.25%   |
> | Unseen           | 87.20±0.98%  | 84.33±2.62%  | 76.50±1.47%   |
> | Total            | 79.88±1.16%  | 76.78±2.28%  | 74.89±1.38%   |
>
> As evident from the results, our method demonstrates only a slight decrease in performance when tested on these more challenging datasets. This minimal degradation in effectiveness underscores our approach's generalizability and adaptability to diverse scenarios, including ambiguous and cross-lingual instructions.

---

> ### Author Response · Authors · 2024-11-20
> **Response to Weaknesses 3, and Questions 1-3**
>
> **W3**: We have indeed considered recent LLM-based RL approaches in our related work section, which can generally be categorized into two types:
>
> 1.Methods like LLaRP[3] and TWOSOME[4] use RL to fine-tune LLMs to output pre-written macro-actions, such as moving to a specific location or grasping an object. However, for complex environments like GRF, defining such macro-actions is challenging. For instance, creating a "past the defender" action in football requires considering the positions and velocities of the player, the ball, and opponents, as well as anticipating opponent reactions. Moreover, in complex environments like GRF 5v5 scenario, training a policy requires tens of millions of samples, making it prohibitively expensive to fine-tune LLMs directly with these samples.
>
> 2.Methods such as SayCan[5] utilize LLMs as high-level planners to select appropriate lower-level RL policies. These approaches also necessitate explicit skill definitions for the lower-level RL policies and require human annotation to determine the success of instruction execution for training. They are also limited to a finite set of behaviors (e.g., 551 in SayCan's case). Consequently, these methods are not easily applicable to complex environments.
>
> To the best of our knowledge, existing LLM-based RL methods are not well-suited for executing instructions in complex environments like GRF. While direct comparisons with other LLM-based RL methods in our complex environments are not feasible due to their limitations, we believe our results provide strong evidence of its effectiveness and uniqueness in instruction alignment for complex RL tasks.
>
> **Additional** : We have created a new anonymous project page that showcases our policy's capabilities through videos. The page can be found at: https://lcmsp-webpage.github.io/LCMSP/.
>
> **Q1**：Please see response to W1.
>
> **Q2**：Please see response to W2.
>
> **Q3**：Please see response to W3.
>
>
> **Reference**
>
> [1] Felix Hill, Sona Mokra, Nathaniel Wong, and Tim Harley. Human instruction-following with deep reinforcement learning via transfer-learning from text. arXiv preprint arXiv:2005.09382, 2020.
>
> [2] Danish Pruthi, Bhuwan Dhingra, and Zachary C Lipton. Combating adversarial misspellings with robust word recognition. arXiv preprint arXiv:1905.11268, 2019.
>
> [3] Andrew Szot, Max Schwarzer, Harsh Agrawal, Bogdan Mazoure, Rin Metcalf, Walter Talbott, Natalie Mackraz, R Devon Hjelm, and Alexander T Toshev. Large language models as generalizable policies for embodied tasks. In The Twelfth International Conference on Learning Representations, 2024.
>
> [4] Weihao Tan, Wentao Zhang, Shanqi Liu, Longtao Zheng, Xinrun Wang, and Bo An. True knowledge comes from practice: Aligning large language models with embodied environments via reinforcement learning. In The Twelfth International Conference on Learning Representations, 2024.
>
> [5]Anthony Brohan, Yevgen Chebotar, Chelsea Finn, Karol Hausman, Alexander Herzog, Daniel Ho, Julian Ibarz, Alex Irpan, Eric Jang, Ryan Julian, et al. Do as i can, not as i say: Grounding language in robotic affordances. In Conference on robot learning, pp. 287–318. PMLR, 2023.

---

### Official Review · Reviewer_3ZBi · 2024-11-02

**Soundness:** 3
**Presentation:** 4
**Contribution:** 2
**Rating:** 5
**Confidence:** 4

**Summary:**

This paper proposes a novel LC-RL method named LCMSP. This method allows one model to be trained with diverse behavioral styles that multi-style parameters can flexibly control.

The trained policy starts with following some instructions and there is no need to evaluate instruction completion during training, which in my option is the major key to achieve multiple personalities/behavior styles for the final RL policy.

LCMSP also leverages LLMs to do evaluation: the language instructions are translated into corresponding multi-style parameters using
a prompting method.

**Strengths:**

Experiments across multiple environments such as highway driving and football playing envs. LCMSP also supports multiple instruction types demonstrating effectiveness and robustness.


This work also propose  Degree-to-Parameter (DTP) prompt with LLMs, which quantifies the goal and style type for the policy. The policy is trained using a specially designed multi-style RL method.

This work sheds light on future studies on efficient ways to integrate natural language comprehension into policy control.

**Weaknesses:**

The demonstration video link doesn't work.

The contribution is not enough. Even though the proposed LCMSP is effective, it relies heavily on the quality of LLMs and policy training. The DTP prompt engineering is a natural consideration.

Need more experiments to prove the generalizability, especially for long-horizon and complicated tasks. Especially for autonomous driving situations, it may require more comprehensive studies.

**Questions:**

See weeknesses

---

> ### Author Response · Authors · 2024-11-21
> **Response to Weaknesses 1-2**
>
> **W1**: We sincerely regret the inconvenience caused by the video link not working. We have provided a new link with the video demonstration included. We hope you can view it and share your feedback. https://lcmsp-webpage.github.io/LCMSP/.
>
> **W2**:  Thanks for your comments. The contributions of our paper can be summarized as below:
>
> **1.Novel Paradigm for Language-Controlled RL**: We have introduced a novel paradigm for controlling RL policies using natural language. Unlike previous language-conditioned RL (LC-RL) methods, which are limited to simple tasks such as object manipulation and navigation, our approach is the first to enable agents to execute highly complex natural language instructions, such as tactics, which encompass abstract concepts, in intricate environments.
> Traditional LC-RL methods typically define tasks represented by instructions and use rule-based judgments (like reward shaping) to guide the training of these specific tasks. In contrast, our approach trains predefined meta-behaviors with different style parameters. By combining various meta-behaviors and style parameters, our method can accomplish instructions that are challenging to guide using rule-based judgments. For example, in a football game, we might want a strategy that embodies "Counter Attack." For traditional LC-RL methods, it's challenging to evaluate how well the agent performs the "Counter Attack" instruction in order to provide rewards for training. However, our method automatically learns various complex behaviors during training by combining meta-behaviors, as shown in Figure 5 and Figure 6.
>
> **2.Multi-Style RL Training Based on Meta-Behaviors**: One of the key contributions of our work is the multi-style RL training framework based on meta-behaviors. Previous multi-style RL approaches, such as [1], have only been tested in simple grid environments, while [2] and [3] were limited to basic game environments like Mario and Sonic, and with a very limited number of styles. For the first time, we have extended multi-style training methods to complex environments like GRF, which involve multiple agents, cooperation and competition, and long horizons. This showcases a wide variety of policies that can be controlled by natural language and resemble real-world tactics. Our work demonstrates the feasibility and effectiveness of multi-style RL methods in such scenarios, significantly expanding the application boundaries of multi-style RL, which previous research has not achieved.
>
> **3.The Effectiveness of the DTP method**:  Our method's effectiveness is not solely dependent on the quality of LLMs. Rather, it derives significant value from the Degree-to-Parameter (DTP) approach, which aligns the LLM with RL policy. To illustrate this, we conducted an experiment comparing the performance with and without the DTP method. In our experiment,  we compared prompts that directly output style parameters without using DTP to generate style parameters for a "Counter Attack" tactic. We then used the trained multi-style policy with these newly generated parameters against the same opponents mentioned in the experiment of Figure 5. Tweleve in-game metrics of these two comprason methods are shown in Table R1 and R2.
>
> Table R1: In-game metrics for Counter Attack tactic with DTP prompt.
>
> | Methods | Win Rate | Tie Rate | Goal Count | Lose Goal Count | Shot Attampts | Pass Attampts | Own Ball Ratio | Get Possession Times | Lose Possession Times | Spacing | Formation |
> |---|----|------|-----|------|-----|----|---|---|---|----|-----|
> | GPT-4o with DTP | 0.412 | 0.247 | 3.962 | 1.062 | 20.200 | 123.002 | 0.319 | 14.740 | 14.068 | 0.364 | 0.355 |
> | Claude3.5-sonnet with DTP | 0.442 | 0.238 | 3.728 | 1.087 | 20.207 | 118.145 | 0.330 | 15.240 | 14.453 | 0.364 | 0.362 |
> | Llama3.1-8b with DTP | 0.457 | 0.257 | 3.717 | 1.010 | 19.683 | 123.130 | 0.331 | 15.180 | 14.290 | 0.360 | 0.363 |
> | Mean ± Std | 0.44 ± 0.02 | 0.25 ± 0.01 | 3.80 ± 0.14 | 1.05 ± 0.04 | 20.03 ± 0.30 | 121.43 ± 2.84 | 0.33 ± 0.01 | 15.05 ± 0.27 | 14.27 ± 0.19 | 0.36 ± 0.00 | 0.36 ± 0.00 |
>
> Table R2: In-game metrics for Counter Attack tactic without DTP prompt.
>
> | Methods | Win Rate | Tie Rate | Goal Count | Lose Goal Count | Shot Attampts | Pass Attampts | Own Ball Ratio | Get Possession Times | Lose Possession Times | Spacing | Formation |
> |-|-|-|----|--------|---------|---------|---------|---------|---------|-------|-------|
> | GPT-4o without DTP | 0.497 | 0.207 | 3.520 | 0.965 | 25.133 | 145.232 | 0.323 | 19.305 | 18.300 | 0.321 | 0.455 |
> | Claude3.5-sonnect without DTP | 0.457 | 0.324 | 3.348 | 0.925 | 21.830 | 146.345 | 0.314 | 18.980 | 18.030 | 0.339 | 0.431 |
> | Llama3.1-8b without DTP | 0.598 | 0.151 | 4.525 | 2.450 | 32.720 | 122.355 | 0.296 | 18.710 | 17.655 | 0.300 | 0.491 |
> | Mean ± Std | 0.52 ± 0.07 | 0.23 ± 0.09 | 3.80 ± 0.64 | 1.45 ± 0.87 | 26.56 ± 5.58 | 137.98 ± 13.54 | 0.31 ± 0.01 | 19.00 ± 0.30 | 18.00 ± 0.32 | 0.32 ± 0.02 | 0.46 ± 0.03 |

---

> > ### Author Response · Authors · 2024-11-21
> > **Response to Weaknesses 2-3**
> >
> > When the DTP method is removed, relying solely on LLMs to generate multi-style parameters, even state-of-the-art LLMs struggle to produce outputs that consistently align with the intended effects of the instructions. Counter-attacking typically requires a more defensive positioning (smaller formation), fewer shots, and less frequent possession changes. However, without the DTP method, the style parameters converted by LLMs directly **fail to exhibit the corresponding style**. Moreover, the results obtained by methods without DTP **vary considerably** across different LLMs. Conversely, the metrics produced by the DTP method aligned more closely with the "Counter Attack" instruction's description and **performed similarly** across different LLMs. These findings underscore the DTP method's capacity to:
> >
> >   ● Foster consistency in judgments across different LLMs.
> >
> >   ● Effectively align the LLM with the RL policy.
> >
> > In summary, DTP serves as a crucial component, enhancing robustness and consistency while ensuring better alignment between instructions and learned behaviors across various LLMs.
> >
> > **4. Comprehensive Testing**: We conducted extensive experiments, testing our method with instructions of varying meanings across different environments. Our results demonstrate that our approach can understand abstract and high-level instructions and their application contexts in a few-shot setting, executing reasonable strategies. We also tested various instruction types, proving the generalizability of our method.
> >
> >
> > **W3**: The 5v5 scenario in GRF is an open-source environment designed for long-horizon and complicated tasks. Each episode consists of 3,000 steps and features sparse rewards. It also incorporates rules such as offside, free kicks, throw-ins, and goal kicks as in realistic football. The task requires simultaneous control of four agents and demands intricate multi-agent cooperation. As an adversarial task, it also necessitates generalizability to compete against opponents with varying styles, making rational decisions based on the current score and remaining time. Under our multi-style training mechanism, our AI has learned a wide array of sophisticated behaviors. These include time-wasting tactics in the defensive area when the match is nearing end, executing wing breakthroughs followed by crosses,  consecutive through balls in the central area combined with off-side trap breaking movements. These behaviors often require execution over hundreds of continuous time steps and necessitate coordination among multiple agents.
> >
> > To simulate real football match scenarios, we designed six instruction sets that represent tactics commonly used in actual football matches, such as park the bus, counter attack, and Tiki-Taka. We believe that the 5v5 scenario in GRF meets the requirements for long-horizon and complex tasks and can demonstrate the generalizability of our method. Importantly, experiments conducted in the 5v5 scenario in GRF are easily reproducible by other researchers. Additionally, we used the Highway environment because it is a lightweight and popular RL training scenario, serving as a more diverse environment to validate the generalizability of LCMSP. The complexity of environments is primarily demonstrated through GRF 5v5.
> >
> > **Reference**
> >
> > [1] de Woillemont Pierre Le Pelletier, Remi Labory, and Vincent Corruble. Configurable agent with reward as input: A play-style continuum generation. In 2021 IEEE Conference on Games (CoG). IEEE, August 2021. doi: 10.1109/cog52621.2021.9619127.
> >
> > [2]Siddharth Mysore, George Cheng, Yunqi Zhao, Kate Saenko, and Meng Wu. Multi-critic actor learning: Teaching RL policies to act with style. In International Conference on Learning Representations, 2022.
> >
> > [3]Runzhe Yang, Xingyuan Sun, and Karthik Narasimhan. A generalized algorithm for multi-objective reinforcement learning and policy adaptation. Advances in neural information processing systems, 32, 2019.

---

### Official Review · Reviewer_U4gW · 2024-11-04

**Soundness:** 3
**Presentation:** 4
**Contribution:** 2
**Rating:** 6
**Confidence:** 4

**Summary:**

This paper proposes a new language-conditioned RL approach called LCMSP (Language-Conditioned Multi-Style Policies) to enable RL models to learn multi-style policies that can be controlled through natural language instructions. To achieve this, LCMSP first identifies and designs a set of meta-behaviors. And together with their corresponding style parameters, these meta-behaviors are used to shape the reward function of RL agents during training. The authors also propose to use a new method called Degree-to-Parameter prompt (DTP) to convert natural language instructions into style parameters, which enables fine-grained control over RL policies through natural language instructions. Then in the experiment section, the authors did comprehensive experiments and analysis in the Highway environment and the GRF environment to show that LCMSP is able to follow abstract language instructions to learn different styles of policy in RL environments.

**Strengths:**

- **Originality**: This paper’s approach is pretty innovative in that it uses meta-behaviors and style parameters to perform targeted reward shaping over RL models’ reward functions in order to control the styles of learned policies. This new methodology could potentially be inspiring for following works in the research field of language-conditioned reinforcement learning.

- **Quality**: The quality of the technical component of this paper is generally good. Most of the key procedures in their proposed LCMSP approach are well supported by corresponding mathematical formulae and descriptions of technical details. Please refer to the Weaknesses section in this review for things to improve.

- **Clarity**: The presentation of this paper is of high quality. The motivation, methodology and empirical analysis of their proposed LCMSP approach are all conveyed clearly in the paper. In their experiment section and the appendix, the authors also conducted very comprehensive experiments to fully evaluate LCMSP’s RL performance in comparison with the baseline method. Many details of their proposed prompting and training procedures are carefully documented in the appendix with tables and figures, which are very helpful for readers to understand and replicate their proposed method.

- **Significance**: The significance of this paper’s core contribution is okay, but not excellent. Please see the Weaknesses section in this review for a detailed discussion on why.

**Weaknesses:**

1. In this paper, the proposed LCMSP method lacks a systematic approach for determining the set of meta-behaviors for a generic RL environment/task. The high-level discussion of determining meta-behaviors in Section 4.1 is a little too vague and general. Although Appendix A.3 provides the specific designs of meta-behaviors for the Highway and the GRF environments, such designs seem a little too ad hoc and too specific to the two environments, and give little insight on how readers can determine the set of meta-behaviors for new unseen RL environments.

2. Ideally, for an RL method to claim that it achieves language-controlled multi-style policies, especially for a method like LCMSP that employs numerical style parameters to enable fine-grained control of policies, after the reinforcement learning process finishes each numerical style parameters should be directly tunable (like a knob or button) to control of style/behaviors of the learned RL policy without the need of retraining. However, for LCMSP, everytime we change the style parameters (evenly slightly), the RL model will need to be trained completely again in the environment to internalize this new set of style parameters from scratch, which is very inefficient. In the current paper manuscript, there is no evidence indicating whether these style parameters in LCMSP have the nice property of ‘smooth linear interpolation for generalization’. That is to say, under LCMSP after the RL model has been trained using a set of style parameters, such as [0.9, 0.8, 0.6], if we now fix all the other parameters of the RL model’s neural architecture and only change the style parameters to [0.1, 0.2, 0.3], will the policy now exhibit style/behaviors such that the corresponding three meta-behaviors all reducing their intensity/frequency? There is currently no evidence in the paper that discusses this important topic. If the answer is no, then this could potentially be an important limitation for the applicability of the paper’s proposed new method and undermines the significance of the paper’s contribution. If the authors have conducted such investigation/analysis, I would encourage the authors to present them in the paper to help readers better understand the properties of LCMSP.

**Questions:**

1. In Section 5.1, why are there no baseline methods being compared with in the experiments for the Highway environment?
2. For the proposed Degree-to-Parameter (DTP) prompting method that converts language instructions into a set of numerical/boolean style parameters, is the DTP conversion process a ‘lossy’ projection or ‘lossless’ projection? If it’s a ‘lossy’ projection, then how can we improve the design process of meta-behaviors in order to minimize the information loss in this DTP conversion process?
3. In general, for the reward calculation during RL training under LCMSP, how can one determine whether the current RL model has successfully ‘triggered/exhibited’ a specific meta-behavior, especially when such meta-behavior is a bit abstract or high level?

---

> ### Author Response · Authors · 2024-11-19
> **Response to Weaknesses 1-2**
>
> **W1**: Thank you for your feedback. We acknowledge that designing meta-behaviors for an environment is a heuristic process that often requires a certain level of understanding of the environment or task itself. During the exploration process of this study, we have summarized two design principles as guidance:
>
> (1) Use as few meta-behaviors as possible to cover the widest range of agent behaviors within the environment.
>
> (2) Ensure that meta-behaviors facilitate easy reward shaping.
>
> This  guidance is akin to the "Occam's Razor" principle, where we aim to describe the majority of an AI agent's behaviors using fewer and simpler elements. This can be achieved through various combinations and degree of these elements, aligning with the concept of multi-style policy training based on meta-behaviors.
>
> We recognize that this requires a certain degree of understanding of the environment, similar to the process of reward shaping in RL. This process can be likened to "labeling" in supervised learning and is essential for the AI study.
>
>
> **W2**: Thank you for your valuable feedback. We think your question can be summarized into two topics: the first concerns the generalization of the trained style parameters, and the second concerns the data efficiency of the method. We will address each of these important issues separately.
>
> (1) Our poposed method exhibits excellent generalization for different style parameters, as demonstrated in Appendix D.4.2 and Figure 13. Figure 13 shows the in-game metrics as functions of the style parameters. It can be observed that adjusting the value of each style parameter results in a nearly linear and smooth change in the corresponding metric. The style parameters tested in the figure may not have been trained, yet the results show strong consistency in game metrics as the style parameters change. This indicates that LCMSP possesses the property of 'smooth linear interpolation for generalization.' Furthermore, although we only sampled style parameters within the range [0, 1], we found that setting a style parameter to 1.1 resulted in further changes in the related game metrics. This suggests that our method not only has 'interpolation generalization' but also some 'extrapolation generalization' capability. We appreciate the reviewer's suggestion and have decided to include the results and discussion of these findings in the next version of our manuscript.
>
> (2) Our approach does not involve training a new set of style parameters from scratch each time. Instead, the style generator simultaneously produces multiple different style parameters, allowing the agent to generate new training data based on these parameters. Once the amount of training data is sufficient for a batch, we perform a training iteration using data from various style parameters. Through multiple training iterations, when the change in training metrics (such as episode returns) stabilizes, we consider that the different style parameters within the sampling range have been effectively captured.
>
> We conducted an experiment to preliminarily demonstrate the efficiency of this approach. The training environment involved drawing shapes like triangles, quadrilaterals, and circles, with three meta-behaviors and corresponding style parameters: one controlling the shape, another controlling the radius, and the third controlling the angular velocity. The specific environment setup can be referenced in [1] section 5.1. Four experimental conditions were established :
>
> Experiment A: One group trained with one set of fixed style parameters.
>
> Experiment B: Three groups, each trained by varying one style parameter while keeping the other two fixed.
>
> Experiment C: Three groups, each trained by varying two style parameters while keeping the other one fixed.
>
> Experiment D: One group trained by varying all three style parameters.
>
> We recorded the episode return curves as the number of used samples increased, with three runs per group using different random seeds. The results showed that, with the same number of samples, the episode return for models in Experiment D was only slightly lower than models in Experiment A and was comparable to models in Experiment B and C. The results indicate that as the number of training styles increases, there is a modest decrease in sample efficiency, though the decline is not severe.This training curve can be found in the last part of our project demonstration: https://lcmsp-webpage.github.io/LCMSP/. These results indicate that the LCMSP method is training efficient.
>
> [1] Mysore, Siddharth, et al. "Multi-critic actor learning: Teaching RL policies to act with style." International Conference on Learning Representations. 2022.

---

> > ### Author Response · Authors · 2024-11-19
> > **Response to Questions 1-3**
> >
> > **Q1**: In the initial design of our experiments, we aimed to focus on different aspects across the different scenarios. The Highway environment was primarily intended to test the basic capabilities of the method, such as alignment accuracy, execution success rates, and the performance of various style parameters. We agree with the reviewer that more comparative experiments should be conducted in the Highway environment. Therefore, we have added two additional comparative experiments: alignment accuracy and execution success rates across different instruction types in the Highway environment.
> >
> > Table R1: Alignment accucary of LCMSP vs. TALAR  on five types of instructions within highway environment.
> >
> > | Instruction Type | LCMSP-gpt4o       | LCMSP-claude3.5-sonnet | LCMSP-llama3.1-8B | TALAR         |
> > |------------------|-------------------|------------------------|-------------------|---------------|
> > | Normal           | **97.4 ± 0.3%**       | 94.6 ± 0.5%            | 95.6 ± 0.4%       | 59.0 ± 5.1%   |
> > | Long             | 83.0 ± 0.5%       | 86.3 ± 1.1%            | **87.3 ± 0.5%**       | 56.3 ± 1.9%   |
> > | Short            | 96.2 ± 0.3%       | **97.2 ± 0.6%**            | 86.7 ± 0.2%       | 60.0 ± 3.7%   |
> > | Inference        | **73.3 ± 3.3%**       | 62.6 ± 2.1%            | 67.8 ± 0.8%       | 22.2 ± 3.9%   |
> > | Unseen           | **97.8 ± 1.1%**       | 92.2 ± 0.7%            | 89.8 ± 0.2%       | 6.7 ± 1.9%    |
> > | Total            | **91.4 ± 0.6%**       | 90.4 ± 0.9%            | 87.3% ± 0.6%      | 49.9 ± 0.9%   |
> >
> > Table R2: Execution rate of TALAR, LCMSP-gpt4o, and Ground truth sytle parameterson five types of instructions within highway environment. (The Normal, Long, and Short type instructions have same Ground truth sytle parameters)
> >
> > | Instruction Type | TALAR            | LCMSP-gpt4o      | Ground Truth style parameters |
> > |------------------|------------------|------------------|-------------------------------|
> > | Normal           | 54.2% ± 1.8%     | 91.5% ± 1.6%     | **92.55% ± 1.9%**                 |
> > | Long             | 56.4% ± 2.6%     | 92.4% ± 2.1%     | **92.55% ± 1.9%**                 |
> > | Short            | 53.5% ± 2.9%     | 88.3% ± 3.7%     | **92.55% ± 1.9%**                 |
> > | Inference        | 25.1% ± 3.9%     | 77.8% ± 3.5%     | **91.32% ± 2.1%**                 |
> > | Unseen           | 5.7% ± 2.1%      | 81.9% ± 3.2%     | **86.03% ± 3.2%**                 |
> > | Total            | 47.06% ± 2.5%    | 88.8% ± 0.7%     | **91.66% ± 2.3%**                 |
> >
> > **Q2**: Thank you for your insightful question. This is indeed a profound topic. For boolean style parameters, the DTP conversion process is a 'lossless' projection. However, for numerical style parameters, it is a 'lossy' projection. This 'lossiness' arises from two main sources: the difference between user intent and user instructions, and the precision loss due to the discretization of the degree represented by the instructions in the DTP process.
> >
> > To minimize information loss in the DTP conversion process from the perspective of designing meta-behaviors, we recommend increasing the proportion of meta-behaviors represented by boolean style parameters, as the conversion of Bool-type meta-behaviors to style parameters is lossless. For numerical style meta-behaviors, if the degrees of meta-behaviors can be naturally described by users in numerical terms, such as "one-third of the force" or "80% of the maximum speed," the conversion process will have less loss. This is because LLMs can easily interpret the numerical values represented by such language, although this condition is quite idealistic.
> >
> > We acknowledge that minimizing information loss in the DTP conversion process is challenging due to the inherent ambiguity and inconsistency in users' understanding and description of language and degrees. Therefore, in our subsequent improvements, we have incorporated an understanding of user context instructions and included the previous style parameters as a reference. This allows users to adjust their instructions, thereby enhancing the system's usability.
> >
> > **Q3**：As we mentioned in our response to W1, the design of meta-behaviors should adhere to the principle of being "easy to perform reward shaping" and should aim to use fewer and simpler meta-behaviors. These meta-behaviors can then be combined in various ways to describe the range of policies an agent can perform in the environment. This concept is somewhat analogous to chemical elements; while there are 118 known chemical elements, they can combine to form millions of different substances. Therefore, we do not recommend designing abstract or high-level meta-behaviors. Instead, consider whether a behavior can be further decomposed into a combination of other simple behaviors. While this approach is idealistic, it is a valuable perspective to adopt in practical engineering applications.

---

### Official Review · Reviewer_XCmQ · 2024-11-04

**Soundness:** 3
**Presentation:** 3
**Contribution:** 1
**Rating:** 3
**Confidence:** 4

**Summary:**

The paper introduced an approach to prompt LLM for changing style parameters during complex games, the style parameters are then used for generating low-level RL policy. The prompt template is evaluated in two environments across 3 models, GPT-4o, Claude-3.5 Sonnet and LLama 3.1-8B, and is proved to be better than BERT-based style parameter translators.

**Strengths:**

- The method seems to achieve the state-of-the-art performance on Multi-Style Reinforcement Learning, though I'm not sure if the baseline is not strong enough, as TALOR is not designed with style parameters.
- The prompt template to generate style parameters is potentially useful to other researchers.
- The presentation of the paper is easy to follow.

**Weaknesses:**

- My major concern is the contribution of the paper. Its novelty only lies in the prompt template, the rest are the same multi-style RL framework from the literatures.
- The paper didn't report latency, which is important as you're testing on autonomous driving tasks and online game playing. I understand the paper does not focus on intermediate language input. But to make it a real-use instead of just an artificial setup, you can imagine a human providing language input for agents while driving or in the middle of the game. In both tasks, in reality we won't prompt LLM for 30s generation as it's enough time for accident in driving or for the opponent to score goals. Instead, a fast response-first (e.g. BERT) followed by LLM prediction overwrite might be the choice.
- In the Highway environment, I like the results on a separated alignment score and execution success. But why TALAR baseline cannot be compared to here? To understand the effectiveness of LCMSP, it would be better if the results could be organized in the following way: a table on LCMSP vs. TALAR on *only* the style parameters prediction accuracy, showing the performance gain from LLM; a table on LCMSP vs. TALAR vs. oracle (GT style parameters) on success rate, showing the gain from a better style parameters and the gap with oracle. The current table is hard for me to understand the contribution and limitation of the proposed method.
- Similar for GRF, it's better to have a result table/figure on the style parameters prediction accuracy.

**Questions:**

From Figure 4, the performance of LLama is close to GPT-4o and Claude-3.5 (you didn't report the overall but I mean roughly), which is not common as other in language understanding task. This means the style parameter prediction task may not require much language understanding or reasoning ability, and makes me wonder why even smaller LM cannot do it. BERT is not the best model to compare because it's encoder only and the text generation definitely requires a good decoder, maybe you can compare to T5-large or GPT-2 (finetuned).

P.S. The citation of TALAR is wrong, it's from Neurips 2023.

---

> ### Author Response · Authors · 2024-11-19
> **Response to Weaknesses 1-2**
>
> W1: Thank you for your comments. The contributions of our paper can be summarized in the following four points:
>
> 1. We have introduced a novel paradigm for controlling reinforcement learning (RL) policies using natural language. Unlike previous language-conditioned RL (LC-RL) methods, which are limited to simple tasks such as object manipulation and navigation, our approach is the first to enable agents to execute highly complex natural language instructions, such as tactics, which encompass abstract concepts, in intricate environments.
>
> 2. Traditional LC-RL methods typically define tasks represented by instructions and use rule-based judgments to guide the training of these specific tasks. In contrast, our approach trains predefined meta-behaviors with different style parameters. By combining various meta-behaviors and style parameters, our method can accomplish instructions that are challenging to guide using rule-based judgments. For example, in a football game, we might want a strategy that embodies "Counter Attack." For traditional LC-RL methods, it's challenging to evaluate how well the agent performs the "Counter Attack" instruction in order to provide rewards for training. However, our method automatically learns various complex behaviors during training by combining meta-behaviors, as shown in Figure 5 and Figure 6.
>
> 3. Previous multi-style RL approaches, such as [1], have only been tested in simple grid environments, while [2] and [3] were limited to basic game environments like Mario and Sonic, and with a very limited number of styles. For the first time, we have extended multi-style training methods to complex environments like GRF, which involve multiple agents, cooperation and competition, and long horizons. This showcases a wide variety of policies that can be controlled by natural language and resemble real-world tactics. Our work demonstrates the feasibility and effectiveness of multi-style RL methods in such scenarios, significantly expanding the application boundaries of multi-style RL, which previous research has not achieved.
>
> 4. We conducted extensive experiments, testing our method with instructions of varying meanings across different environments. Our results demonstrate that our approach can understand abstract and high-level instructions and their application contexts in a few-shot setting, executing reasonable strategies. We also tested various instruction types, proving the generalizability of our method.
>
> W2: Thanks for your suggestion.
>
> 1. To eliminate network latency interference caused by calling closed-source LLMs, we tested the inference time of open-source LLMs of different sizes under various scenarios and instruction sets, deployed on our own servers. For Llama3.2-1B, Qwen2.5-0.5B, and BERT, we used an Nvidia A10 GPU for inference, while for the larger Llama3.1-8B, we used an Nvidia L40 GPU. As shown in the Table R1, the inference times for LLMs in the single-player and two-player scenarios are fast. In the most complex 5v5 environment, to reduce inference latency, we optimized the Chain of Thought (CoT) in the prompt, requiring about 4 seconds of inference time, while still ensuring tactical style parameters that conform to the instruction.
>
> Table R1: Inference times of different LLMs and BERT.
> | Scenario                        | Llama3.1-8B       | Llama3.2-1B       | Qwen2.5-0.5B     | BERT              |
> |---------------------------------|-------------------|-------------------|------------------|-------------------|
> | Single-player and Two-player in GRF| 2.78 ± 0.43    | 0.77 ± 0.09       | 0.46 ± 0.07      | 0.013 ± 0.0002    |
> | 5v5 in GRF                      | 4.02 ± 0.26       | 1.29 ± 0.17       | 1.02 ± 0.24      | None              |
> | Highway                         | 1.57 ± 0.57       | 0.62 ± 0.08       | 0.22 ± 0.1       | 0.012 ± 0.0002    |
>
> 2. In practical applications, the multi-style policy and the alignment process of instructions run in parallel, which helps maintain user experience to some extent. Additionally, this method is more suited for scenarios where a strategy needs to be sustained over a period, rather than those requiring immediate responses. Therefore, the impact of latency on user experience is minimal. For example, in a football game, a coach issues a tactical instruction and observes its effectiveness over time before making adjustments.
>
> 3. In our previous demo video, part of the observed delay was due to network latency from API calls rather than actual inference latency. We have re-recorded a version using open-source LLMs deployed on our own servers with low inference latency for demonstration, which can be found in: https://lcmsp-webpage.github.io/LCMSP/

---

> > ### Author Response · Authors · 2024-11-19
> > **Response to Weaknesses 3-4**
> >
> > W3: Thank you for your constructive feedback. In response, we have conducted two new experiments in the Highway environment based on your suggestions. The first experiment compares the alignment accuracy (style parameter prediction accuracy) of different methods, with the results presented in Table R2. The second experiment evaluates the execution success of LCMSP-GPT4o, TALAR, and ground truth style parameters, with the results shown in Table R3. We hope these additional experiments enhance the content of the paper. The implementation details of TALAR are described in Appendix D.3.
> >
> > Table R2: Alignment accucary of LCMSP vs. TALAR of all baselines on five types of instructions within highway environment.
> >
> > | Instruction Type | LCMSP-gpt4o       | LCMSP-claude3.5-sonnet | LCMSP-llama3.1-8B | TALAR         |
> > |------------------|-------------------|------------------------|-------------------|---------------|
> > | Normal           | 97.4 ± 0.3%       | 94.6 ± 0.5%            | 95.6 ± 0.4%       | 59.0 ± 5.1%   |
> > | Long             | 83.0 ± 0.5%       | 86.3 ± 1.1%            | 87.3 ± 0.5%       | 56.3 ± 1.9%   |
> > | Short            | 96.2 ± 0.3%       | 97.2 ± 0.6%            | 86.7 ± 0.2%       | 60.0 ± 3.7%   |
> > | Inference        | 73.3 ± 3.3%       | 62.6 ± 2.1%            | 67.8 ± 0.8%       | 22.2 ± 3.9%   |
> > | Unseen           | 97.8 ± 1.1%       | 92.2 ± 0.7%            | 89.8 ± 0.2%       | 6.7 ± 1.9%    |
> > | Total            | 91.4 ± 0.6%       | 90.4 ± 0.9%            | 87.3% ± 0.6%      | 49.9 ± 0.9%   |
> >
> > Table R3: Execution rate of TALAR，LCMSP-gpt4o, and ground truth sytle parameters on five types of instructions within highway environment. (The Normal, Long, and Short type instructions have same Ground Truth sytle parameters)
> >
> > | Instruction Type | TALAR            | LCMSP-gpt4o      | Ground truth style parameters |
> > |------------------|------------------|------------------|-------------------------------|
> > | Normal           | 54.2 ± 1.8%     | 91.5 ± 1.6%     | 92.55 ± 1.9%                 |
> > | Long             | 56.4 ± 2.6%     | 92.4 ± 2.1%     | 92.55 ± 1.9%                 |
> > | Short            | 53.5 ± 2.9%     | 88.3 ± 3.7%     | 92.55 ± 1.9%                 |
> > | Inference        | 25.1 ± 3.9%     | 77.8 ± 3.5%     | 91.32 ± 2.1%                 |
> > | Unseen           | 5.7 ± 2.1%      | 81.9 ± 3.2%     | 86.03 ± 3.2%                 |
> > | Total            | 47.06 ± 2.5%    | 88.8 ± 0.7%     | 91.66 ± 2.3%                 |
> >
> > W4: Thank you for your comments. The results for the style parameters prediction accuracy in the GRF environment are already provided in Appendix D.3 and Figure 11.

---

> ### Author Response · Authors · 2024-11-19
> **Response to Questions 1-2**
>
> Q1: Thank you for your insightful comments. To ensure fairness, we selected two of the latest language models with parameter sizes comparable to T5-large and GPT-2 for testing. The comparason results are shown in Table R4 and Table R5. In the GRF single-player and two-player scenarios with simple instructions, the performance of the 8B open-source LLM (Llama3.1-8B) is close to that of the best closed-source LLMs (GPT-4o). However, smaller language models tend to show a decline in performance. This decline is partly due to alignment errors and partly because smaller LLMs are more prone to generating responses in incorrect formats, which cannot be parsed into style parameters. The BERT model, after training, achieves a performance level similar to the few-shot capabilities of a 1B parameter LLM (Llama3.2-1B). Given BERT's rapid inference time, methods based on BERT, such as TALAR, are superior for scenarios requiring quick decision-making.
>
> Table R4: Alignment accucary of six baselines on five types of instructions within GRF Single-player scenario
>
> |Instruction Type| TALAR|LCMSP-Qwen2.5-0.5B|LCMSP-Llama3.2-1B|LCMSP-GPT-4o|LCMSP-Claude3.5-Sonnet|LCMSP-Llama3.1-8B|
> |--------------|--------------|----------------|---------------|-------------|--------------------|---------------|
> | Normal   | 66.47 ± 1.11% | 33.67 ± 0.94%  | 63.83 ± 1.31% | 83.13 ± 0.62% | **88.70 ± 1.06%**     | 84.62 ± 1.39%   |
> | Long     | 41.27 ± 2.22% | 36.50 ± 0.71%  | 65.17 ± 2.71% | **91.27 ± 1.73%** | 83.60 ± 1.96%     | 91.08 ± 1.48%   |
> | Short    | 57.93 ± 1.31% | 25.67 ± 0.47%  | 59.17 ± 4.36% | 69.07 ± 1.23% | **69.30 ± 1.57%**     | 60.63 ± 2.94%   |
> | Inference| 43.20 ± 10.04%| 17.83 ± 3.01%  | 38.00 ± 0.33% | **58.07 ± 1.00%** | 30.25 ± 2.07%     | 29.52 ± 1.91%   |
> | Unseen   | 58.00 ± 2.83% | 31.83 ± 0.62%  | 49.17 ± 0.84% | **87.20 ± 0.98%** | 85.40 ± 2.20%     | 72.94 ± 1.39%   |
> | Total   | 54.52 ± 2.36% | 30.80 ± 0.88% | 59.42 ± 2.65%  | **79.88 ± 1.16%** | 76.95 ± 1.64%   | 74.24 ± 1.88%     |
>
> Table R5: Alignment accucary of six baselines on five types of instructions within GRF Two-player scenario
>
> | Instruction Type | TALAR          | LCMSP-Qwen2.5-0.5B | LCMSP-Llama3.2-1B | LCMSP-GPT-4o     | LCMSP-Claude3.5-Sonnet | LCMSP-Llama3.1-8B |
> |------------------|----------------|--------------------|-------------------|-----------------|------------------------|-------------------|
> | Normal           | 71.85 ± 4.93%    | 32.00 ± 0.40%       | 59.33 ± 0.94%     | 93.00 ± 0.28%     | **95.65 ± 0.55%**            | 87.30±0.64%       |
> | Long             | 68.90 ± 6.34%    | 26.17 ± 0.47%      | 55.17 ± 2.89%     | 81.80 ± 1.35%     | **85.00 ± 1.36%**            | 84.20±0.73%       |
> | Short            | 72.65 ± 4.58%    | 35.33 ± 1.84%      | 54.00 ± 1.63%     | 82.80 ± 0.79%     | 86.30 ± 0.82%            | **99.70±0.30%**       |
> | Inference        | **67.40 ± 6.12%**    | 1.83 ± 0.04%      | 20.17 ± 2.01%     | 44.45 ± 0.96%     | 51.40 ± 1.33%            | 55.05±0.96%       |
> | Unseen           | 37.90 ± 4.50%    | 31.50 ± 2.27%      | 49.50 ± 2.94%     | 79.65 ± 1.94%     | **85.65 ± 1.35%**            | 82.75±1.03%       |
> | Total            | 67.62 ± 5.28%    | 28.83 ± 1.00%      | 52.56 ± 1.94%     | 81.93 ± 0.93%     | 85.63 ± 0.99%            | **86.81±0.64%**       |
>
> Q2:  Thank you for your meticulous review. We will correct the citation for TALAR to reflect its publication in NeurIPS 2023.
>
> **Reference**
>
> [1] de Woillemont Pierre Le Pelletier, Remi Labory, and Vincent Corruble. Configurable agent with reward as input: A play-style continuum generation. In 2021 IEEE Conference on Games (CoG). IEEE, August 2021. doi: 10.1109/cog52621.2021.9619127.
>
> [2]Siddharth Mysore, George Cheng, Yunqi Zhao, Kate Saenko, and Meng Wu. Multi-critic actor learning: Teaching RL policies to act with style. In International Conference on Learning Representations, 2022.
>
> [3]Runzhe Yang, Xingyuan Sun, and Karthik Narasimhan. A generalized algorithm for multi-objective reinforcement learning and policy adaptation. Advances in neural information processing systems, 32, 2019.

---

### Meta-Review · Area_Chair_RWWJ · 2024-12-21

**Metareview:**

The paper introduced an approach to prompt LLM for changing style parameters during complex games, the style parameters are then used for generating low-level RL policy. The prompt template is evaluated in two environments across 3 models, GPT-4o, Claude-3.5 Sonnet and LLama 3.1-8B, and is proved to be better than BERT-based style parameter translators.

Overall in terms of strengths, the paper clearly illustrates the efficacy of the approach. However, reviewers had concerns about the overall contribution of the paper, with several reviewers noting that approach was rather incremental, with limited contribution.

**Additional Comments On Reviewer Discussion:**

The reviewers and authors had a productive discussion. Overall, however all reviewers remained unconvinced about the merits of the paper.

---

### Decision · Program_Chairs · 2025-01-22

Reject